# Wintertime direct radiative effects due to black carbon (BC) over Indo-Gangetic Plain as modelled with new BC emission inventories in CHIMERE

Sanhita Ghosh[1], Shubha Verma[2], Jayanarayanan Kuttippurath[3], and Laurent Menut[4]

[1]Advanced Technology Development Centre, Indian Institute of Technology Kharagpur, Kharagpur-721302, India
[2]Department of Civil Engineering, Indian Institute of Technology Kharagpur, Kharagpur-721302, India
[3]Centre for Oceans, Rivers, Atmosphere and Land Sciences (CORAL), Indian Institute of Technology Kharagpur, Kharagpur-721302, India
[4]Laboratoire de Météorologie Dynamique, IPSL, CNRS / Ecole Polytechnique / Sorbonne Université / Ecole Normale Supérieure, 91128 Palaiseau Cedex, France

**Correspondence:** Shubha Verma (shubha@iitkgp.ac.in)

**Abstract.** To reduce the uncertainty in the black carbon (BC) induced climatic impacts from the global and regional aerosol-climate model simulations, it is a foremost requirement to improve the prediction of modelled BC distribution. And that specifically, over the regions where the atmosphere is loaded with a large amount of BC, e.g., the Indo-Gangetic plain (IGP) in the Indian subcontinent. Here we examine the wintertime direct radiative perturbation due to BC with an efficiently modelled BC distribution over the IGP in a high-resolution ($0.1° \times 0.1°$) chemical transport model, CHIMERE, implementing new BC emission inventories. The model efficiency in simulating the observed BC distribution was assessed executing five simulations: $Constrained$ and $bottomup$ ($Smog$, $Cmip$, $Edgar$, $Pku$) implementing respectively, the recently estimated India-based constrained BC emission and the latest bottom-up BC emissions (India-based: Smog-India, and global: Coupled Model Intercomparison Project phase 6 (CMIP6), Emission Database for Global Atmospheric Research-V4 (EDGAR-V4) and Peking University BC Inventory (PKU)). The mean BC emission flux from the five BC emission inventory database was found to be considerably high (450–1000 $\mathrm{kg\,km^{-2}\,yr^{-1}}$) over most of the IGP, with this being the highest (>2500 $\mathrm{kg\,km^{-2}\,yr^{-1}}$) over the megacities (Kolkata and Delhi). A low estimated value of the normalised mean bias (NMB) and root mean square error (RMSE) from $Constrained$ estimated BC concentration (NMB: < 17%) and aerosol optical depth due to BC (BC-AOD) (NMB: 11%) indicated that simulation with constrained BC emissions in CHIMERE could simulate the distribution of BC pollution over the IGP more efficiently than with the bottom-up. The large BC pollution covering the IGP region comprised of wintertime all-day (daytime) mean BC concentration and BC-AOD, respectively, in the range 14–25 (6–8) $\mathrm{\mu g\,m^{-3}}$ and 0.04–0.08 from the $Constrained$. The simulated BC concentration and BC-AOD were inferred to be primarily sensitive to the change in BC emission strength over most of the IGP (including megacity Kolkata), but also that to the transport of BC aerosols over megacity Delhi. Five main hotspot locations were identified in and around Delhi (northern-IGP), Prayagraj/Allahabad-Varanasi (central-IGP), Patna-Palamu (upper/ lower mideastern-IGP), and Kolkata (eastern-IGP). The wintertime direct radiative perturbation due to BC aerosols from the $Constrained$ comprised of the atmospheric radiative warming ($+30$ to $+50$ $\mathrm{W\,m^{-2}}$)

to be about 50%–70% larger than the surface cooling. A wide-spread enhancement in atmospheric radiative warming due to BC by 2-3 times and a reduction in surface cooling by 10%-20%, with net warming at the top of atmosphere (TOA) of 10-15 W m$^{-2}$ was noticed, compared to the atmosphere without BC, for which, a net cooling at the TOA was, although, exhibited. These perturbations were spotted being the strongest around megacities (Kolkata and Delhi) extended to the eastern coast and were inferred as 30%–50% lower from the $bottomup$ than the $Constrained$.

## 1  Introduction

Black carbon (BC) is released into the atmosphere from the incomplete combustion of carbon-based fuels (Bond et al., 2013; Verma et al., 2013; Sadavarte and Venkataraman, 2014). It is one of the constituents of concern among the atmospheric aerosol pollutants because of its profound impact on climate through an imbalance of the Earth's radiation budget, besides, degradation of air quality and adverse effects on human health as well (Qian et al., 2011; Wang et al., 2014a; Fan et al., 2015; Zhang et al., 2015; Janssen et al., 2011, 2012). Among aerosol constituents, BC aerosols are considered the strongest absorber of visible solar radiation and, thereby, a prominent contributor to tropospheric warming as for the greenhouse gases– carbon dioxide and methane (Ramanathan and Carmichael, 2008; Gustafsson and Ramanathan, 2016; Masson-Delmotte et al., 2018). However, the magnitude of tropospheric radiative warming due to BC aerosols is highly uncertain and is classified with a medium to low-level understanding in the Inter-governmental Panel on Climate Change–Fifth Assessment Report (IPCC–AR5) (Myhre et al., 2013a, b; Wang et al., 2016; Boucher et al., 2016; Permadi et al., 2018a; Paulot et al., 2018; Dong et al., 2019). The direct radiative forcing (DRF) of BC averaged over the globe is estimated in the range 0.2–1 W m$^{-2}$ (Myhre et al., 2013b; Bond et al., 2013; Gustafsson and Ramanathan, 2016). These estimates from global climate models used in the latest assessment by the IPCC is noted to be about 2-times lower than the observation–based estimates from satellite and ground-based Aerosol Robotic Network (AERONET) observations (0.7–0.9 W m$^{-2}$) (Chung et al., 2012; Myhre et al., 2013b; Gustafsson and Ramanathan, 2016; Stocker et al., 2014). The DRF of BC is inferred to be furthermore uncertain (e.g., $-0.06$ W m$^{-2}$ to $+0.22$ W m$^{-2}$) when estimated for BC rich sources comprising of BC emitted with different composition of short-lived co-emissions of species, e.g., sulphate and organic carbon (Bond et al., 2013).

Though the consensus is still to be achieved in BC DRF, nevertheless, the global atmospheric absorption attributable to BC was found to be too low in models and had to be enhanced by a factor of three to converge with observation-based estimates (Bond et al., 2013). The systematic underestimation of BC aerosol absorption by the global climate model predictions relative to atmospheric observations as noticed specifically over south Asia and east Asia (Chung et al., 2012; Gustafsson and Ramanathan, 2016) is also in compliance with studies evaluating atmospheric BC concentration between model and observations. For example, recent evaluations of BC concentration from global and regional aerosol models over south Asia showed that the simulated BC concentration, though, exhibited a consistent correlation with, but was significantly lower (by a factor of about 2 to 11) than the measured concentration (Kumar et al., 2018; Verma et al., 2017; Kumar et al., 2015; Pan et al., 2015; Sanap et al., 2014; Moorthy et al., 2013; Nair et al., 2012). The factor of model underestimation was further noticed to be large

specifically during wintertime over the Indo-Gangetic Plain (IGP) when the atmosphere is observed to be laden with a large BC burden.

To assess BC aerosol absorption accurately and reduce the uncertainty in the BC DRF as estimated from global and regional aerosol-climate models, it is, therefore, a foremost requirement to improve the prediction of atmospheric BC estimates in models. And that specifically, over the regions where the atmosphere is loaded with a large amount of BC, e.g., the Indo-Gangetic plain (IGP) in the Indian subcontinent (Nair et al., 2007; Verma et al., 2013; Ram and Sarin, 2015; Thamban et al., 2017; Rana et al., 2019). Possible reasons suggested for the discrepancy between model and observations included, lack of BC emissions used as input, inadequate meteorology, and representation of aerosol treatment, and coarse resolution in the model (e.g. Santra et al., 2019; Kumar et al., 2018; Wang et al., 2016; Pan et al., 2015; Verma et al., 2011; Reddy et al., 2004).

However, it is also noted from the evaluation of BC concentration estimated from the free-running aerosol simulations using Laboratoire de Météorologie Dynamique atmospheric general circulation model (LMDZT-GCM) that simulated BC which is underestimated by a significant factor at stations which are close to emission sources (such as that over mainland India), exhibit a relatively lower discrepancy with observed BC concentration over the Indian oceanic regions (Reddy et al., 2004; Verma et al., 2007, 2011). The simulated BC distribution from LMDZT-GCM was also found to match consistently well the available observations at high altitude Himalayan Hindukush stations (e.g., Hanle, Satopanth) which are relatively remotely located and mostly influenced by the transport of aerosols (Santra et al., 2019). The above evaluations, therefore, suggest that the large underestimation of BC concentration over the India mainland would primarily be due to BC emission dataset, instead of the model configurations.

The simulated atmospheric BC burden with atmospheric chemical transport models is related to the BC emission strength as input and simulated atmospheric residence time of BC (Textor et al., 2006). While the atmospheric residence time of BC aerosols is independent of the emission strength, it is an indication of model-specific treatments of transport and aerosol processes affecting the simulated BC burden. The uncertainty in the mean model residence time for BC based on evaluation in sixteen global aerosol models, has been estimated as 33% (Textor et al., 2006), which is, though, noted to be much lower than the discrepancy found between the simulated BC and observation. Due to the inclusion of various complex physical-chemical atmospheric and aerosol processes in these models, in conjunction with the inherent uncertainty in inputs to the model (e.g., aerosol emissions and their properties), a systematic approach is required to improve the prediction of BC aerosols in the models. The uncertainty in bottom-up BC emission inventory has been inferred about greater than 200% over India and Asia (Bond et al., 2004; Streets et al., 2003; Venkataraman et al., 2005; Lu et al., 2011), compared to that about 40% in recently estimated constrained BC emission over India (Verma et al., 2017). Therefore, in the above context, it is required to assess the efficacy of simulating the BC burden in a state-of-the-art chemical transport model under the different emission scenarios (e.g., bottom-up and constrained) evaluating the divergence in BC emission flux from state-of-the-art bottom-up BC emission inventories and constrained BC emission.

In this study, we examine the wintertime direct radiative effects of BC over the IGP evaluating the efficacy of simulated atmospheric BC burden in a high resolution (0.1°×0.1°) chemical transport model, CHIMERE, during winter when a large BC burden is observed. This is done executing multiple BC transport simulations with CHIMERE, implementing new BC

emission inventories, which included the recently estimated India-based constrained BC emissions and the latest bottom-up BC emissions (India-based: Speciated Multi-pOllutant Generator (Smog-India), and global: Coupled Model Intercomparison Project phase 6 (CMIP6), Emission Database for Global Atmospheric Research-V4 (EDGAR-V4) and Peking University BC Inventory (PKU)). A short description of the five BC emission datasets is provided in Section 2.1. The bottom-up BC emissions applied in the present study are being widely used in regional and global climate models in the assessment of spatial and temporal distribution of aerosol burden and aerosol-climate interactions (Eyring et al., 2016; Zhou et al., 2020; David et al., 2018; Lamarque et al., 2010; Meng et al., 2018; Wang et al., 2016), including (e.g., CMIP6) to support the IPCC climate assessment report (Myhre et al., 2013a). Henceforth, it is necessary to evaluate the performance of the new BC emissions (bottom-up and constrained), with a state-of-the-art chemical transport model, towards their adequacy to represent the BC distribution and thereby, the climatic impacts, over the IGP in the Indian subcontinent. The model efficiency in simulating the observed BC distribution, including the spatial and temporal trend, is, thus, examined with the estimated BC concentration from five simulations subjected to the same aerosol physical and chemical processes with CHIMERE. Besides, the surface BC concentration, which is observed to be large during winter compared to summer (Pani and Verma, 2014) owing potentially to a wintertime shallow planetary boundary layer height (PBLH) (also discussed in Section 3.1). It is also necessary to evaluate the wintertime columnar BC loading (Chen et al., 2020), which has implications for BC radiative perturbations. To assess the columnar distribution of BC aerosols, aerosol optical depth due to BC (BC-AOD) and its fractional contribution to total AOD, are also examined, in conjunction with presenting an analysis on the wintertime radiative perturbation due to BC aerosols. Note that applications presented in this paper focus on the aerosol–radiation interactions only and show the wintertime direct radiative perturbations or the direct radiative effects (DRE) due to BC. The study of indirect aerosol effects referring to cloud–aerosol interactions, evaluating changes in the number of cloud condensation nuclei, including the perturbations of the cloud albedo and rainfall (Boucher et al., 2013; Lohmann and Feichter, 2005) is currently ongoing and shall be presented in a forthcoming study.

The specific objectives of this study are, therefore, to (i) characterise the model efficiency from five simulations through a detailed validation and statistical analysis of simulated BC concentration with respect to ground-based measurements at stations over the IGP, and identify the regional hotspots, (ii) utilise the multi-simulations to quantify the degree of variance in estimated BC concentration attributed to emissions corresponding to areas types (e.g., megacity, urban, semi-urban, low-polluted) and temporal distribution (e.g., daytime and evening hours), (iii) evaluate the spatial features of BC-AOD from five simulations, and analyse the association between simulated BC concentration and BC-AOD with BC emissions strength, and (iv) examine the spatial distribution of wintertime radiative perturbation due to BC aerosols over the IGP and that compared with the atmosphere considered without BC aerosols.

## 2 Method of study

### 2.1 Experimental set-up for simulating BC surface concentration

High-resolution BC transport simulations are carried out with a state-of-the-art Eulerian chemical transport model (CTM), CHIMERE. The CHIMERE (model version 2014b) configuration in the present study is forced externally by Weather Research and Forecasting (WRF-V3.7) model as a meteorological driver in offline mode, meaning that the meteorology is pre-calculated with WRF then read in CHIMERE. Further, to compute the radiative perturbations due to BC, an offline coupling is executed again, forcing the WRF model with aerosol optical properties computed from CHIMERE output (refer to Section 2.3). Thereby implying the need to incorporate interactions between the two models using a WRF-CHIMERE online coupled modelling system for computing aerosol-radiation-cloud interactions (Briant et al., 2017; Péré et al., 2011). Simulations are carried out at a horizontal grid resolution of $0.1° \times 0.1°$ and over the domain spanning from 20°N to 30.8°N and 75°E to 89.9°E including IGP region. BC transport simulations are performed for the winter of December 2015, keeping a spin up time of 15-days in November 2015, from 15 to 30 November. Evaluation of atmospheric BC concentration and BC-AOD in the present study is done during the winter month of December when the winter season is well developed in India and when the monthly mean of BC concentration is typically observed being the highest (e.g., Pani and Verma (2014)). Simulation is done for the year 2015 as the recent bottom-up BC emission database over India (Smog-India) as implemented in the present study is for the year 2015.

### 2.1.1 The CHIMERE chemical transport model

CHIMERE is a regional chemical transport model designed to model ten number of gaseous species and aerosols. For chemistry, the gaseous mechanism MELCHIOR2 is used (Derognat et al., 2003). The calculation of aerosols is as described in Bessagnet et al. (2004) with ten bins, with a mean mass median distribution ranging from 0.039 to 40 $\mu$m and for primary particulate matter (black carbon, BC, organic carbon, OC, and PPMr–the remaining part of primary emissions), sulphate, nitrate, ammonium, sea salt, and water. Biogenic, dust, and sea salt emissions are calculated online within CHIMERE. Biogenic emissions are estimated with the model of Emissions of Gases and Aerosols from Nature (Guenther et al., 2006). Mineral dust and sea salt emissions are parameterized following Menut et al. (2015) and Monahan (1986), respectively. Secondary organic aerosols are formed following Bessagnet et al. (2009). Chemical concentration fields are calculated with a time-step of few minutes (using an adaptive time-step sensitive to the mean wind speed). For radiation and photolysis, the online FastJX model is used (Wild et al., 2000). The horizontal transport is calculated with the VanLeer scheme (van Leer, 1979) and vertical using an upwind scheme with mass conservation Menut et al. (2013). Note that additional information is provided in Table 1 (bottom). Boundary layer height is diagnosed using the Troen and Mahrt (1986) scheme, and deep convection fluxes are calculated using the Tiedtke (1989) scheme. Gaseous and aerosol species can be dry or wet deposited, and fluxes are computed using the Wesely (1989); Zhang et al. (2001) parameterizations. Initial and boundary conditions are estimated using global model monthly climatology calculated with the Laboratoire de Météorologie Dynamique General Circulation Model coupled with Interaction with Chemistry and Aerosols (LMDz-INCA) (Szopa et al., 2009). The domain grid has twenty vertical levels in $\sigma$-pressure coordinates ranging from the surface (997 hPa) to 200 hPa. CHIMERE reads the WRF hourly meteorological fields

and interpolates these meteorological fields if the CHIMERE grid is different. The interpolation is a bilinear interpolation, ensuring mass conservation for variables needing it. CHIMERE also reads anthropogenic emissions fields. The user can use the CHIMERE dedicated program (called EMISURF, see Menut et al. (2012)) or make its own program and create a file on the CHIMERE grid directly.

### 2.1.2   The WRF meteorological model

The WRF model is a state-of-the-art numerical weather forecast and atmospheric simulation system designed for both research and operational applications. The initial and boundary meteorological conditions for WRF simulation are obtained from Global Forecast System (GFS) National Center for Environmental Prediction - FINAL operational global analysis data (NCEP-FNL, http://rda.ucar.edu/datasets/ds083.2/) at a spatial resolution of $1° \times 1°$. Meteorological fields are simulated in WRF at the temporal resolution of one-hour with the horizontal resolution same as that for CHIMERE simulation. The meteorological boundary conditions are updated every six hours. The optimized schemes applied in WRF simulation are as follows: Lin scheme for cloud microphysics (Lin et al., 1983), Grell 3D ensemble scheme for subgrid convection (Grell and Devenyi, 2002), Yonsei university (YSU) scheme for boundary layer (Hong et al., 2006), Rapid Radiative Transfer Model (RRTM) for radiation transfer (Mlawer et al., 1997), MM5 Monin-Obukhov scheme for surface layer and Noah LSM for land-surface model (Chen and Dudhia, 2001).

### 2.1.3   Implementation of BC emissions and multiple CHIMERE simulations

In the present study, five simulations are carried out subjected to the same model processes with CHIMERE but implementing different BC emission inventories. The BC inventories include recently estimated India-based– (i) constrained and (ii) bottom-up BC emissions (Smog-India), including the bottom-up BC emissions from global datasets extracted over India– (iii) EDGAR-V4 (EDGAR), (iv) CMIP6 and (v) PKU. Spatially and temporally resolved gridded constrained BC emission over India is taken as per Verma et al. (2017). The observationally-constrained BC emissions or so-called constrained BC emissions were estimated using integrated forward and receptor modelling approaches (Kumar et al., 2018; Verma et al., 2017). The estimation was done extracting information on initial bottom-up BC emissions and atmospheric BC concentration from the general circulation model (Laboratoire de Météorologie Dynamique atmospheric General Circulation Model (LMDZT-GCM)) simulation. The receptor modelling approach involved estimating the spatial distribution of potential emission source fields of BC based on mapping the concentration weighted trajectory (CWT) fields of measured BC (daytime averaged) corresponding to the identified stations over the Indian region. The constrained BC emissions were obtained, modifying the initial or baseline bottom-up BC emissions of the GCM corresponding to the emission source fields of BC, constraining the simulated BC concentration in the GCM with the observed BC (refer to Verma et al. (2017) for formulation and details).

BC emission inventory based on bottom-up approach is generally compiled using information on activity data and generalised emission factors (see the references for bottom-up emissions, Table 1 (top)). The recent bottom-up BC emission database over India implemented is from Smog-India (Pandey et al., 2014; Sadavarte and Venkataraman, 2014). The CMIP6 BC emission used in the model simulations of CMIP6 is a combination of regional and global emission inventories and re-gridded as

per EDGAR-V4 (Eyring et al., 2016). In the present study, global BC emission inventories utilised, $viz.$ emission Database for EDGAR, CMIP6, and PKU are re-gridded to the resolution as per the Smog-India database. The BC transport simulation in CHIMERE corresponding to emission database- Constrained, Smog-India, EDGAR, CMIP6, PKU are referred to as, respectively, $Constrained$, $Smog$, $Edgar$, $Cmip$ and $Pku$. The annual BC emission strength over the study domain as estimated from the implemented inventories lies in the range 415–1517 Gg yr$^{-1}$. Details of simulation experiment and a short description of BC emission inventories implemented are summarised in Table 1 (top). The classified source sectors of BC emission from the emission inventory database include residential, open burning, energy and industry, and transportation. The annual BC emission strength corresponding to each of the source sectors as available from the emission inventory database is also mentioned in Table 1 (top). The fuel combustion activity among the source sectors includes the combustion of fuelwood, crop-waste, dung-cake, kerosene, and cooking-LPG for residential cooking and heating corresponding to the 'residential' sector; open burning of agricultural residue, grassland, trash, and forest biomass to 'open burning' source sector; coal, and diesel for energy to 'energy and industry' sector; and diesel, petrol, gasoline to the 'transportation' sector. Based on the available information on sector-wise BC emission source strength (Table 1 (top)), the residential sector is seen to be the largest contributor to BC emission over the Indian region consistent with Venkataraman et al. (2005). The magnitude of annual BC emission source strength corresponding to all the sectors except the 'energy and industry' sector is estimated to be 2- to 3-times larger for the constrained emission than the bottom-up. This is specifically larger for the open burning sector, noted as 3-times the bottom-up Smog-India, thereby suggesting the specific improvement required in quantifying the BC emission strength of the open burning sector in the bottom-up BC emission inventory. Interestingly, compared to the rest of the other source sectors, the BC emission source strength of the 'energy and industry' sector from the constrained emission matches relatively well with that from the bottom-up Smog-India. The seasonality in the spatial and temporal distribution of BC emission strength is inferred mainly from the open burning sector due to region- and season-specific prevalence of open burning of the crop residues after harvesting of Rabi or Kharif crops, including that of forest biomass burning over the Indian subcontinent (Venkataraman et al., 2006; Verma et al., 2017). The BC emission flux is also noted as being the largest during winter months over the entire Indian subcontinent and is specifically large over the IGP (Verma et al., 2017).

Besides BC emission, emission of OC, SO$_2$, and PPMr are also implemented in CHIMERE. This implementation is done to perform atmospheric aerosol transport simulation for atmosphere with abundant aerosol species (including BC), and that for atmosphere without BC. These simulations are required to calculate the radiative perturbations due to BC aerosol (refer to Section 2.3). The spatial distribution of mean and percentage standard deviation ($\delta$ as represented in Equation 4) of BC emission flux from five BC emission inventories over the study domain is presented in Figures 1a and 1b, respectively. The mean BC emission flux is considerably high (450–1000 kg km$^{-2}$ yr$^{-1}$) over most of the IGP, with this being the highest (>2500 kg km$^{-2}$ yr$^{-1}$) over the megacities (Kolkata and Delhi). The divergence in BC emission flux is about 50%–75% over most of the IGP with this being relatively lower over the eastern and upper mideastern IGP. The divergence is large in and around megacities (100%–125%), and is noted to be specifically large (150%–200%) over the rural location in the lower mideastern IGP (in and around Palamu, refer to Figure 3e for details of location). Uncertainties in activity data and emission factors have been inferred leading to uncertainty in bottom-up BC inventories of about greater than 200% over India and Asia, as also

mentioned earlier (Streets et al., 2003; Bond et al., 2004; Venkataraman et al., 2005; Lu et al., 2011). One of the drawbacks of the bottom-up approach is its inability to take into account possible unknown or missing emission sources. Bottom-up BC emissions are thus found to be often lower than the actual (Rypdal et al., 2005; Johnson et al., 2011; Zhang et al., 2005; Reid et al., 2009). Bottom-up BC emission over India includes a large missing source of BC emitted over India (Venkataraman et al., 2006). Hence, the divergence in emission data (refer Figure 1b) using five emission datasets (observationally-constrained and bottom-up BC emissions) is indicative of inadequacy in BC emission source strength suggesting specific improvement required in bottom-up BC emission tabulation over the IGP and that at specific locations where the divergence is typically noted to be large.

**Table 1.** Experimental setup for simulation of BC with CHIMERE

| | | | Description of BC emission inventories implemented | | |
|---|---|---|---|---|---|
| Name of the simulation | Emission (Resolution) | Domain (approach) | Annual BC emission strength over study domain, Gg yr$^{-1}$ (base year) | Source sectors (Emission strength, Gg yr$^{-1}$) | References |
| *Smog* | Smog-India (0.25° × 0.25°) | Indian (bottom-up) | 817 (2015) | Residential (393) Open burning (98) Energy and Industry (163) Transportation (163) | Sadavarte and Venkataraman (2014); Pandey et al. (2014); https://sites.google.com/view/smogindia |
| *Edgar* | EDGAR (0.1° × 0.1°) | Global (bottom-up) | 579 (2010) | Residential (400) Energy and Industry (127) Transportation (52) | Janssens-Maenhout et al. (2012); http://edgar.jrc.ec.europa.eu/ |
| *Cmip* | CMIP6 (0.5° × 0.5°) | Global (bottom-up) | 558 (2014) | Anthropogenic (547) (Residential, energy, industry, Commercial, transportation) Open burning (11) | Eyring et al. (2016); Feng et al. (2020); https://esgf-node.llnl.gov/projects/cmip6/ |
| *Pku* | PKU (0.1° × 0.1°) | Global (bottom-up) | 415 (2007) | Residential Open burning Energy and Industry Transportation | Wang et al. (2014b); http://inventory.pku.edu.in |
| *Constrained* | Constrained (0.25° ×0.25°) | Indian (constrained: integrated receptor modelling with GCM) | 1517 (latest) | Residential (744) Open burning (273) Energy and Industry (212) Transportation (288) | Verma et al. (2017) http://www.facweb.iitkgp.ac.in/~shubhaverma/constrained-bc-emissions-over-India.html |

| | Details of aerosol module of CHIMERE for BC (Menut et al., 2013) |
|---|---|
| Number of bins for BC | 10 (Mass-median diameter interval: 0.039, 0.078, 0.156, 0.312, 0.625, 1.25, 2.5, 5, 10, 20, 40 $\mu$m) |
| Aerosol mixing | Internal homogeneous |
| Aerosol dynamics | Absorption, nucleation, coagulation, aging of BC |
| Deposition | Dry deposition and in cloud or below cloud wet deposition |

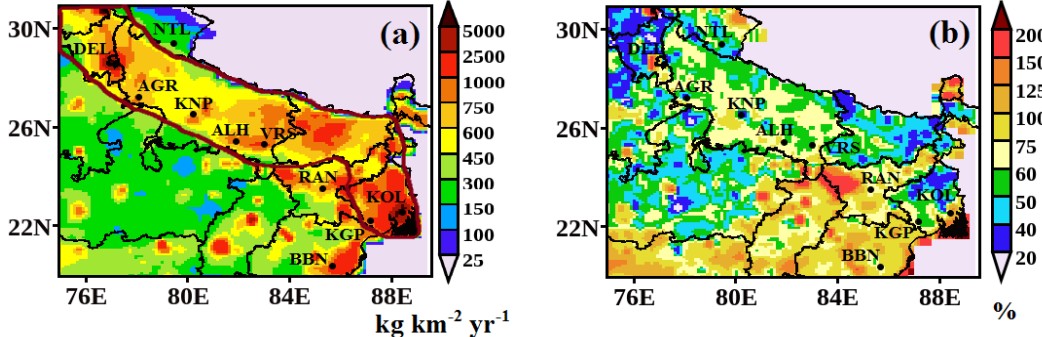

**Figure 1.** Spatial distribution of (a) mean and (b) percentage deviation ($\delta$) of BC emission flux from five BC emission inventories implemented in CHIMERE over the study domain; the brown line in Figure (a) indicates the IGP region.

## 2.2   Observational data for model evaluation and model sensitivity analysis

The spatial distribution of WRF simulated surface temperature over the IGP is compared with the available gridded distribution of observed temperature from Climatic Research Unit (CRU) (Morice et al., 2012). The observed temperature from CRU at a horizontal resolution of $0.5° \times 0.5°$ is re-gridded to the same resolution ($0.1° \times 0.1°$) as that from WRF and the bias in simulated temperature for each grid-cell is calculated using equation 1. The temporal trend of WRF simulated hourly mean of meteorological parameters (temperature, relative humidity) is also evaluated with that of observed from available measurements

at stations over the IGP (Table 2). The monthly mean of simulated PBLH averaged is compared with that of measured available for stations at Delhi (mean of hourly PBLH during 1000–1600 LT), Kharagpur (1000–1100 LT and 1400–1500 LT), Ranchi (at 1430 LT) and Nainital (0500–1000 LT) corresponding to the overlapping time hours from measurements (Figure 2l in Section 3.1). The vertical distribution of the wintertime monthly mean of the potential temperature (Stull, 1988) as obtained from WRF simulated temperature is also compared with the measured vertical distribution of potential temperature for December

(obtained for 2-days) at a station of Kanpur from an available study (Table 2) corresponding to the overlapping time hours (1000–1200 LT) from measurements.

To compare simulated BC surface concentration with observations, measured BC surface concentration is obtained at stations over the IGP from available studies (refer to Table 2 and references therein). The selected stations correspond to area types identified as megacity (Delhi and Kolkata), urban (Agra, Kanpur, Prayagraj (or Allahabad) and Varanasi), semi-urban

(Kharagpur, Ranchi, and Bhubaneshwar) and low polluted (Nainital). Comparing model results with measurements thus aids in fulfilling the requirement to evaluate the model performance towards reproducing the observed spatial patterns in BC distribution for the various area types. Measurement data used in the present study are reported with an uncertainty (due to instrument artifacts, etc.) of about 10%–30% for PBLH (Seidel et al., 2010; Srivastava et al., 2010), 2%–3% for meteorological parameters, and 5%–20% for measured BC concentration measured (refer to Table 2 for details and references therein). It is to be

noted that observational data used for comparing model estimates (meteorological parameters and BC aerosols), belong to measurement during different years at stations over the IGP. The inter-annual variability of PBLH (based on observations over

Delhi) is reported as within 10% (Iyer and Raj, 2013)) and of surface temperature (based on available measurements data of India Meteorological Department over Kolkata and Kharagpur) is less than 6%. The inter-annual variability of atmospheric BC concentration over the Indian subcontinent is obtained as 5%–10% (Safai et al., 2014; Surendran et al., 2013; Bisht et al., 2015; Ram et al., 2010b; Kanawade et al., 2014; Pani and Verma, 2014). Taking into account that the reported inter-annual variability of meteorological parameters and atmospheric BC concentration is nearly equivalent to or within the uncertainty range for measurements and also is much lower than the discrepancy between simulated and observed BC as reported in previous studies (refer to Section 1). The uncertainty range is taken into consideration while evaluating the model performance compared to measurements. The comparison between model estimates and measurements at widespread geographical locations and area types as presented in this study is, therefore, justified and is primarily required for evaluating the model performance and enhancing the statistical analysis.

The model estimated and measured BC concentration are compared corresponding to daytime (1000–1600 LT) and all-day (24-hourly) winter monthly mean values. This comparison is made because measured BC concentrations are found exhibiting a strong diurnal variability, with a relatively lower value during daytime hours than that during the late evening to early morning hours attributed to prevailing wintertime meteorological conditions (Verma et al., 2013; Pani and Verma, 2014). Also, the day-time mean BC concentration exhibits a low hourly variability and corresponds to the well-mixed layer of atmosphere (Verma et al., 2013; Pani and Verma, 2014). Hence, the lower value of the daytime mean from the model than from observations is primarily attributable to a low emission strength. Evaluation of model estimates for both daytime and all-day mean, thus, provides a systematic hypothetical approach to identify the model discrepancy, if primarily due to emissions or that due to model processes attributed to meteorology (which is an input to the various aerosol processes that govern the atmospheric residence time of aerosols). This approach is further strengthened, implementing BC emissions from five new BC emission inventory databases and simulating BC transport subjected to the same aerosol physical and chemical processes with CHIMERE.

Bias in simulated estimates ($X^{modelled}$) from simulations at stations mentioned above for hourly, all-day and daytime hours is estimated with respect to observed data ($X^{obs}$) with the equation as follows,

$$Bias = \frac{(X^{modelled} - X^{obs})}{X^{obs}} \times 100\% \tag{1}$$

where, X = BC concentration, temperature, relative humidity, wind speed and PBLH.

Statistical analyses are carried out corresponding to daytime and all-day winter monthly mean to evaluate the normalised mean bias (NMB, equation 2) and root mean square error (RMSE, equation 3) from the simulated results for N (=10 in this study) number of stations. We also evaluate the percentage deviation ($\delta$) in simulated BC concentration attributed to BC emission, estimated as the variability about the mean of BC concentration from five simulations (refer to equation 4).

Further, the BC-AOD estimated in the present study (refer to Section 2.3) is compared with aerosol absorption optical depth (AAOD) from Aerosol Robotic Network (AERONET; level: 2) based measurements over the IGP (Giles et al., 2012; Holben et al., 1998) at stations of Kanpur, New Delhi-IMD, Gandhi College (25.87°N, 84.12°E) and IIT Kharagpur extension at

Kolkata. The wintertime AAOD available from AERONET observations for Kanpur, New Delhi-IMD, Gandhi College (December 2010–2015 averaged) and Kolkata (February 2009 averaged) are used in the comparison. For Kolkata, the comparison is also made with the estimated BC-AOD of the December 2010 period as obtained from the configured aerosol model using in-situ ground-based observations of the same period (Verma et al., 2013). The AERONET AAOD data are available at four wavelengths: 440, 675, 870, and 1020 nm. The AAOD at the wavelength of 550 nm (used for comparison with simulated BC-AOD in the present study) is obtained based on the wavelength dependence of AAOD as per Giles et al. (2012).

A correlation study is also carried out between the variance of emission and simulated BC concentration or simulated BC-AOD from the simulations to examine the sensitivity of simulated BC concentration or BC-AOD towards the variation in emission magnitude.

$$NMB = \frac{\sum_1^N \left| BC^{modelled} - BC^{obs} \right|}{\sum_1^N BC^{obs}} \cdot 100\% \tag{2}$$

$$RMSE = \left[ \frac{1}{N} \sum_1^N (BC^{modelled} - BC^{obs})^2 \right]^{\frac{1}{2}} \tag{3}$$

$$\delta = \frac{\sigma}{mean} \times 100\% \tag{4}$$

where $\sigma$ is the standard deviation for the mean from five simulations (e.g., BC emissions, all-day, daytime mean of BC concentration, etc.).

### 2.3 Simulation of wintertime BC-AOD and radiative perturbations due to BC over the IGP

The BC-AOD is estimated with OPTical properties SIMulation (OPTSIM) (Stromatas et al., 2012) using the 3-dimensional BC mass concentration obtained from CHIMERE corresponding to each of the five simulations (refer to Table 1). Aerosol optical properties are estimated based on Mie theory calculations considering internal mixing (Lesins et al., 2002; Permadi et al., 2018b). These estimations are done at six wavelengths of 440, 500, 532, 550, 870, and 1064 nm and the same horizontal and temporal resolution as of CHIMERE.

For radiative transfer calculations, estimates from $Constrained$ (which is obtained as the most efficient to simulate the BC distribution, as discussed later) and $Smog$ (based on India-based BC emission as a representative $bottomup$) are only considered. For estimating the radiative effect due to BC aerosols, simulation of aerosol optical properties (AOD, single scattering albedo (SSA) and angstrom exponent (AE), etc.) is conducted with OPTSIM for three different cases considering, respectively, (i) atmosphere including BC ('with BC', $BCaero$), (ii) atmosphere without BC ('without BC', $wBC$) and (iii) atmosphere

**Table 2.** Observational data used for model validation from available studies at identified locations over the study domain

| Type | Stations | Location | Data (Year of measurement) | References |
|---|---|---|---|---|
| Megacity | Delhi (DEL) | 28.58°N, 77.20°E | BC conc. (2004) <br> PBLH (2006) <br> AAOD (2011–15) | Ganguly et al. (2006) <br> Bano et al. (2011) <br> https://aeronet.gsfc.nasa.gov |
| | Kolkata (KOL) | 22.54°N, 88.42°E | BC conc., temp., RH, wind speed (2011–14) <br> AAOD (2009) <br> BC-AOD (2009–11) | Pani and Verma (2014); <br> Research group, IIT-KGP <br> https://aeronet.gsfc.nasa.gov <br> Verma et al. (2013) |
| Urban | Agra (AGR) <br> Kanpur (KNP) | 27.20°N, 78.10°E <br> 26.51°N, 80.23°E | BC conc. (2004) <br> BC conc. (2007) <br> Vertical profile of potential temp. (2004) <br> AAOD (2011–15) | Safai et al. (2008) <br> Ram et al. (2010a) <br> Tripathi et al. (2005b) <br><br> https://aeronet.gsfc.nasa.gov |
| | Gandhi College <br> Prayagraj/Allahabad (ALH) <br> Varanasi (VRS) | 25.87°N, 84.12°E <br> 25.41°N, 81.91°E <br> 25.30°N, 83.00°E | AAOD (2013) <br> BC conc. (2004) <br> BC conc. (2009) | https://aeronet.gsfc.nasa.gov <br> Badarinath et al. (2007) <br> Singh et al. (2015) |
| Semi-urban | Kharagpur (KGP) | 22.19°N, 87.19°E | BC conc., temp., RH, wind speed (2011–14) <br> PBLH (2004) | Priyadharshini (2019); <br> Research group, IIT-KGP <br> Nair et al. (2007) |
| | Bhubaneswar (BBN) <br> Ranchi (RAN) | 20.50°N, 85.5°E <br> 23.50°N, 85.30°E | BC conc. (2010–11) <br> BC conc. (2010) <br> PBLH (2011) | Mahapatra et al. (2014) <br> Lipi and Kumar (2014) <br> Chandra et al. (2014) |
| Low polluted | Nainital (NTL) | 29.37°N, 79.45°E | BC conc. (2004–07) <br> PBLH (2011) | Dumka et al. (2010) <br> Singh et al. (2016) |

with no aerosol ('without aerosol', $wAero$). The 3-dimensional aerosol species concentration as an input to OPTSIM is derived for each of the three cases from CHIMERE corresponding to simulations ($Constrained$ and $Smog$).

Aerosol radiative transfer calculations are done in WRF-solar at a temporal resolution of 1 hour and horizontal grid resolution of $0.1° \times 0.1°$ selecting a regular longitude-latitude projection. The WRF Preprocessing System (WPS) internally converts grid resolution corresponding to longitude-latitude projection in the degrees to meters required for model processing. The WRF-solar is a new version of the WRF model enhanced for the prediction of solar irradiance (Haupt et al., 2016; Jimenez et al., 2016). The meteorological initial and boundary conditions provided to the model are as per the WRF model, as mentioned previously (refer to 2.1.2). The Rapid Radiative Transfer Model for Global model scheme (RRTMG) (Iacono et al., 2008) is opted for the shortwave and longwave radiation. The direct and diffused components of solar irradiance are separately

addressed with the RRTMG scheme to improve the model calculations by considering surface irradiance components in the estimation.

Simulation for radiative flux with WRF-solar are performed for each of the three cases, as mentioned above, using respective simulated optical properties as input for each case. Shortwave (SW) radiative flux (at 550 nm) for clear sky condition is estimated at the top (TOA) and bottom (SUR) layer of the atmosphere for atmosphere with BC and that without BC. This is done by subtracting the respective flux at TOA and SUR due to $wAero$ from the flux due to $wBC$ and $BCaero$, respectively. The direct radiative perturbations or the direct radiative effects (DRE) due to BC aerosols at TOA ($DRE^{TOA}$(BC)) and at SUR ($DRE^{SUR}$(BC)), which are calculated by taking difference between the radiative flux from $BCaero$ and that from $wBC$ at the respective layers of the atmosphere (equation 5 and 6). The DRE at the atmosphere (ATM) due to BC is estimated by subtracting the flux at the SUR from that estimated at TOA (equation 7).

$$DRE^{TOA}(BC) = [DRE^{TOA}(BCaero) - DRE^{TOA}(wAero)] - [DRE^{TOA}(wBC) - DRE^{TOA}(wAero)] \tag{5}$$

$$DRE^{SUR}(BC) = [DRE^{SUR}(BCaero) - DRE^{SUR}(wAero)] - [DRE^{SUR}(wBC) - DRE^{SUR}(wAero)] \tag{6}$$

$$DRE^{ATM}(BC) = DRE^{TOA}(BC) - DRE^{SUR}(BC) \tag{7}$$

## 3  Results and discussions

### 3.1  Analysis of WRF simulated meteorological parameters

The WRF simulated winter monthly mean of distribution of the horizontal wind speed, vertical wind velocity, and PBLH over the IGP are presented in Figures 2a-c. As observed from the wind-field distribution map, there is a predominance of the weak north-easterlies (1–2 m s$^{-1}$) over the IGP. The vertical wind velocity distribution indicates a neutral or a downdraft of the air mass over the IGP (positive value of the vertical wind velocity is an indication of downdraft of air mass and vice versa). The presence of narrow PBLH (200 m to 600 m) over most of the IGP indicates a low vertical mixing during winter (Figure 2c). The topographical elevation decreases from the northern IGP towards the eastern IGP, with the maximum elevation observed on the northward side due to the Himalayan mountains (Figure 2d).

High load of BC aerosols over the IGP as obtained (discussed later) in the present study is inferred due to confinement of pollution near the surface within the shallow boundary layer height in winter due to low vertical mixing and weak dispersion of atmospheric pollutants, thereby, stagnant weather under the prevailing meteorological conditions, viz. low temperature and

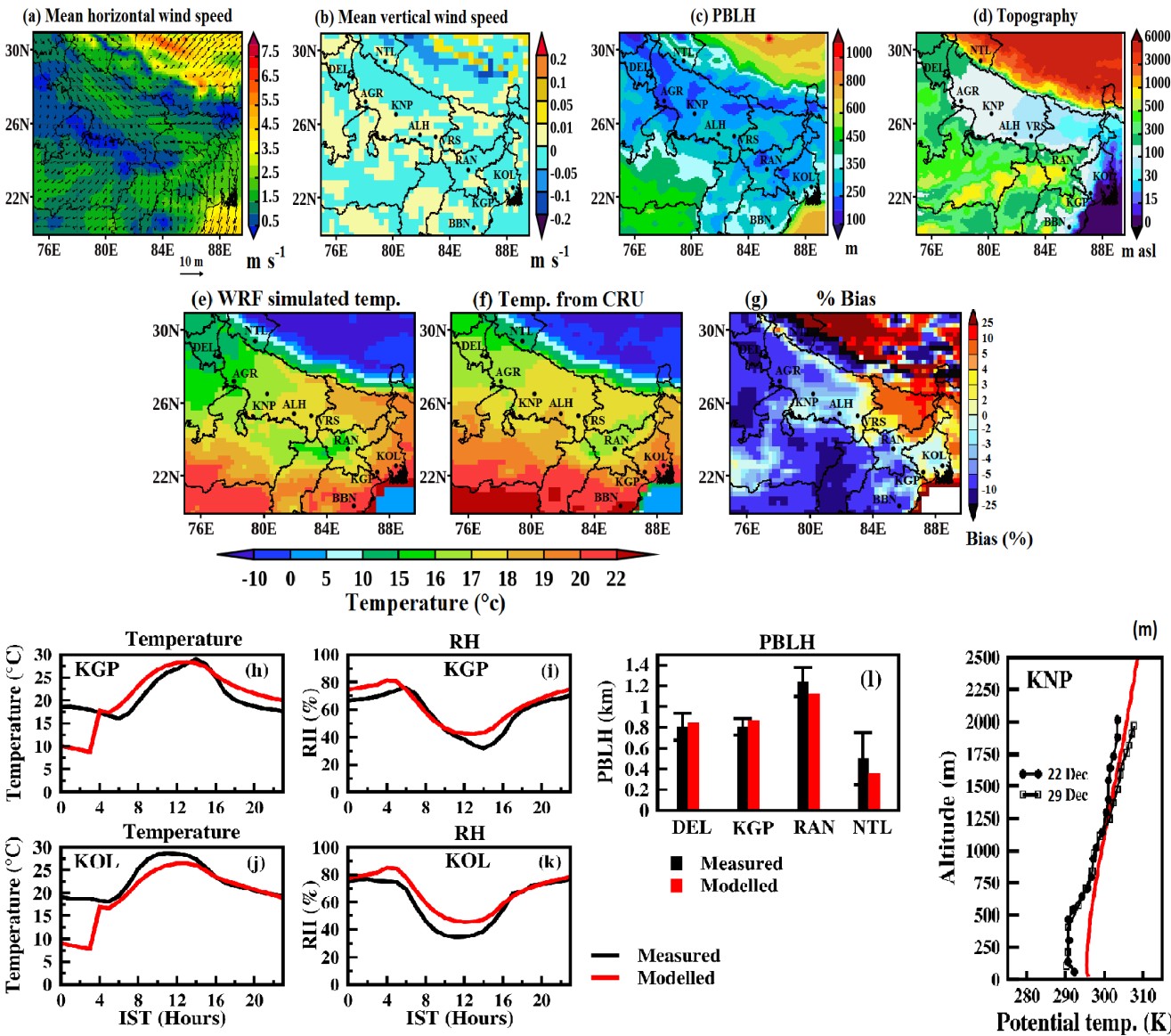

**Figure 2.** Spatial distribution of WRF simulated (a–c) winter monthly mean of (a) horizontal wind field (note the color scale is for wind speed in m s$^{-1}$ and the arrows indicate the direction of the mean field), (b) vertical wind speed at 1000 hPa, (c) planetary boundary layer height (PBLH in m), and (d) topography (m above sea level, m asl); (e–g) spatial distribution of winter monthly mean of surface temperature from (e) WRF simulations, (f) observations from CRU, (g) percentage bias in WRF estimates; (h–k) comparison of hourly distribution of winter monthly mean of (h,j) surface temperature, (i,k) relative humidity from WRF simulations with observations at stations (Kharagpur, KGP; Kolkata, KOL); (l) comparison of winter monthly mean of PBLH during day hours between measured and simulated values at stations under study. The error bars represent the standard deviation ($\sigma$) in measured PBLH; (m) comparison of vertical profile of potential temperature obtained from WRF (winter monthly mean) and measurements (two days).

weak wind speed, the downdraft of the air mass, and a narrow PBLH (as presented above). Besides, the Himalayan mountains

northward, further, inhibits the dispersion of aerosol pollutants and favours their confinement over the IGP. This inference is also in corroboration with the observational studies at stations over the IGP (e.g., Nair et al., 2007, 2012; Pani and Verma, 2014; Verma et al., 2014; Vaishya et al., 2017; Rana et al., 2019). Further, the IGP also comprises the highest population density, and thereby the enhanced BC emission strength during winter, specifically from biofuel combustion, e.g., fuelwood and crop-waste for residential cooking and heating (Venkataraman et al., 2005; Sahu et al., 2015; Verma et al., 2017; Rana et al., 2019).

We compare the spatial distribution of monthly mean temperature from WRF simulations (Figure 2e) with that from gridded ground-based observations from CRU (Figure 2f). The bias in model estimated temperature is found within $\pm 5\%$ over most of the IGP (Figure 2g) but is noticed to be slightly large (about $\pm 10\%$ to $\pm 25\%$) over a few grids of the north-eastern, western and southern IGP. A comparative study of the hourly distribution of winter monthly mean of the simulated surface temperature and relative humidity (RH), with the corresponding observed value from available measurements at Kharagpur (semi-urban)

and Kolkata (megacity) is presented in Figures 2h-k. Please refer to Table 2 for details on observational data. The temporal trend of simulated hourly winter monthly mean of meteorological parameters conform to the measurements. The magnitude of hourly distribution of winter monthly mean of surface temperature from simulations (Figures 2h and 2j) is found to be comparing well with that from observations during daytime hours (1000–1600 LT) for both the stations; but is, however, seen to be underestimated (bias: $-45\%$ to $-58\%$) during mid-night to early morning hours (0000–0500 LT). The WRF simulated

meteorology is input to various aerosol processes that govern the atmospheric residence time of aerosols in CHIMERE, and thereby influences the atmospheric concentration of BC aerosols. A lower value of simulated surface temperature than the observed during mid-night to early morning hours would lead to a decreased mixing of pollutants enhancing their accumulation in the atmosphere during these hours (as also evinced in the diurnal distribution of simulated BC concentration, refer to Section 3.2, Figure 4).

The WRF simulated RH at both stations (Figure 2i and 2k) is in good agreement with measurements (bias: $-5\%$ to $+35\%$) with the mean RH during late evening to early morning hours (2000–0500 LT) being 2-times higher than that during daytime. A comparison of winter monthly mean of PBLH during daytime hours (as described in Section 2.2) from WRF simulation with that available from observations at Delhi, Kharagpur, Ranchi, and Nainital is also presented (Figure 2l). The standard deviations ($1\sigma$) in measured PBLH values are within 10%–16% for Delhi, Kharagpur, and Ranchi and about 49% at Nainital.

The simulated PBLH is close enough to measurements (bias estimated within $\pm 10\%$) at all stations. Although, at Nainital the simulated bias is large ($-28\%$), though, is within the range of uncertainty in observations as mentioned in Section 2.2. The vertical profile of potential temperature (Figure 2m) from WRF (wintertime monthly mean) resembles well with that from measured at a station of Kanpur, with the bias being less than 4% up to the height of 500 m and less than 1% at a higher altitude ($> 500$ m).

Thus, the WRF simulated winter monthly mean of the meteorological parameters, including their temporal trend, conforms well with the observed. However, it is required to reduce the discrepancy, specifically in the simulated magnitude of temperature during mid-night to early morning hours. A better temporally resolved meteorological boundary condition in WRF (compared to 6-hourly from NCEP in the present study), aided with data assimilation at a fine temporal resolution (e.g., 1-hourly) using

diurnal meteorological observations for India-based stations would potentially lead to simulate the observed magnitude of diurnal distribution of meteorological parameters more accurately; an assessment in this regard is required in a future study.

### 3.2 Simulated wintertime BC concentration with new BC emissions as modelled with CHIMERE: impact of changing emissions and comparison with measurements

The spatial distribution of winter monthly mean of BC surface concentration from five simulations over the IGP is shown in Figures 3a-e. Simulated mean BC concentration from $Constrained$ is, in general, 2 to 4 times higher than that derived from $bottomup$ over most of the IGP. Five hotspots or patches (refer to Figure 3e) with large BC concentration (magnitude >16 $\mu$g m$^{-3}$) from $Constrained$ are identified in and around megacities (Delhi and Kolkata) and surrounding semiurban areas, urban spots over central- (Prayagraj/ Allahabad–Varanasi) and mideastern-IGP (Patna), and including the rural spot over the lower mideastern-IGP (Palamu). It is interesting to see that the hotspots observed in $Constrained$ are also identified in $Pku$, mostly in $Smog$ as well, though with a smaller value than the $Constrained$. Estimates from $Edgar$ and $Cmip$ simulate the megacity hotspots but fail to show the rest of the other identified hotspots in $Constrained$. Interestingly, the hotspot at Palamu (a coal mining belt in Jharkhand) is simulated in $Constrained$ and $Pku$, unlike the rest of other simulations, thereby suggesting the lack of BC emission source strength corresponding to Palamu and other identified hotspot locations in bottom-up BC emissions (as also mentioned in Section 2.1.3). The simulated spatial pattern (refer to Figures 3f-g) of BC surface concentration in $Constrained$ while exhibiting the lowest value at high altitude and low-polluted location (e.g., Nainital), and the moderately high values at semi-urban stations (e.g., Kharagpur and Ranchi) to high values at urban stations (e.g., Agra, Kanpur, Allahabad/Prayagraj, Varanasi) is seen to reach the maximum at megacities (Kolkata and Delhi). In comparison to the $Constrained$, the simulated spatial gradient within and across the area types is seen to be inconsistent in $bottomup$ estimates. For example, the $bottomup$ estimated values of BC concentration from $Smog$ and $Cmip$ have a low spatial gradient from megacity (Kolkata) to urban area type, including a lack of spatial contrast compared to the observed among the urban stations; the $Pku$ overestimates the all-day mean BC concentration for megacities but simulates relatively well the spatial gradient across the area types; the $Edgar$ matches well the observed values at megacities for all-day mean but then underestimated the observed by a large value over and with a low spatial contrast among the urban and semi-urban stations.

The simulated spatial pattern in $Constrained$ is consistent with observations (Figures 3f-g) as the BC distribution features for the specific area types are represented well by the simulated BC distribution. The spatial feature also indicates that the wintertime all-day mean value of BC concentration (Figure 3g) at Delhi is lower than Kolkata, and is vice versa for the daytime mean value (Figure 3f); although the BC emission strength (refer to Figure 1a) of the two megacities, Delhi and Kolkata, is nearly equivalent. We discuss these features in context with the transport of BC aerosols over the IGP based on the visualization of an animation (doi: http://doi.org/10.5446/48819, in supplementary material) later in the Section. The simulated magnitude of BC surface concentration from $Constrained$, compared to that from $bottomup$, resembles relatively well with the measured counterpart (Figures 3f-g), with the ratio of measured to simulated all-day (daytime) mean BC concentration being equivalent to nearly one. A detailed statistical analysis of the comparison between simulated and observed BC is presented later in this section.

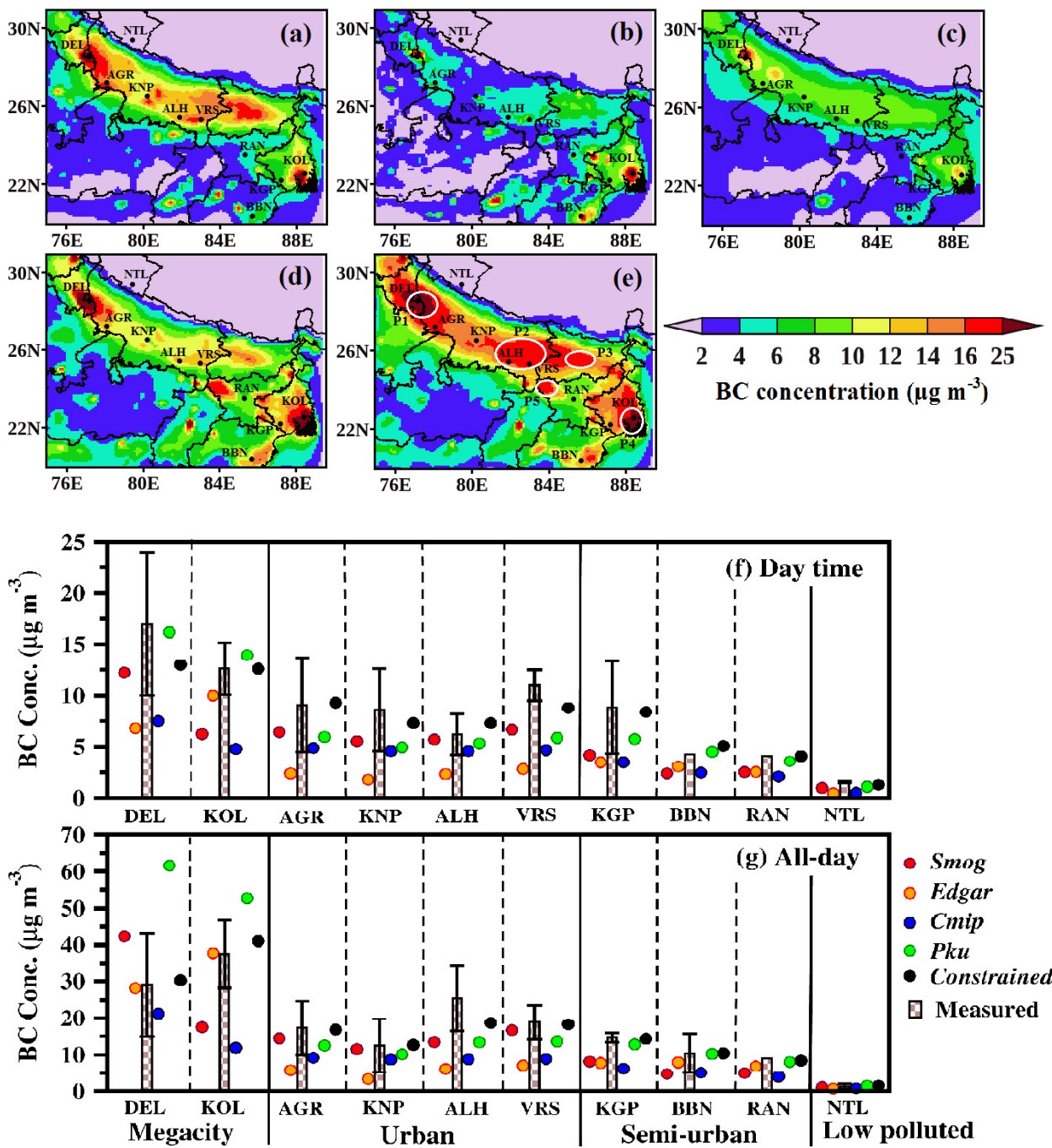

**Figure 3.** (a–e) Spatial distribution of simulated winter monthly mean of BC surface concentration from (a) *Smog*, (b) *Edgar*, (c) *Cmip*, (d) *Pku* and (e) *Constrained*; the circles in white in (e) represent the hotspots with patches of high BC concentrations, P1: Delhi-patch, P2: Prayagraj/Allahabad-Varanasi-patch, P3: Patna-patch, P4: Kolkata-patch, and P5: Palamu-patch;(f–g) comparison of simulated monthly (f) daytime and (g) all-day mean of BC surface concentration from five simulations with measurements at respective stations under study over the IGP; error bars represent the standard deviation (1$\sigma$) in measured BC concentration.

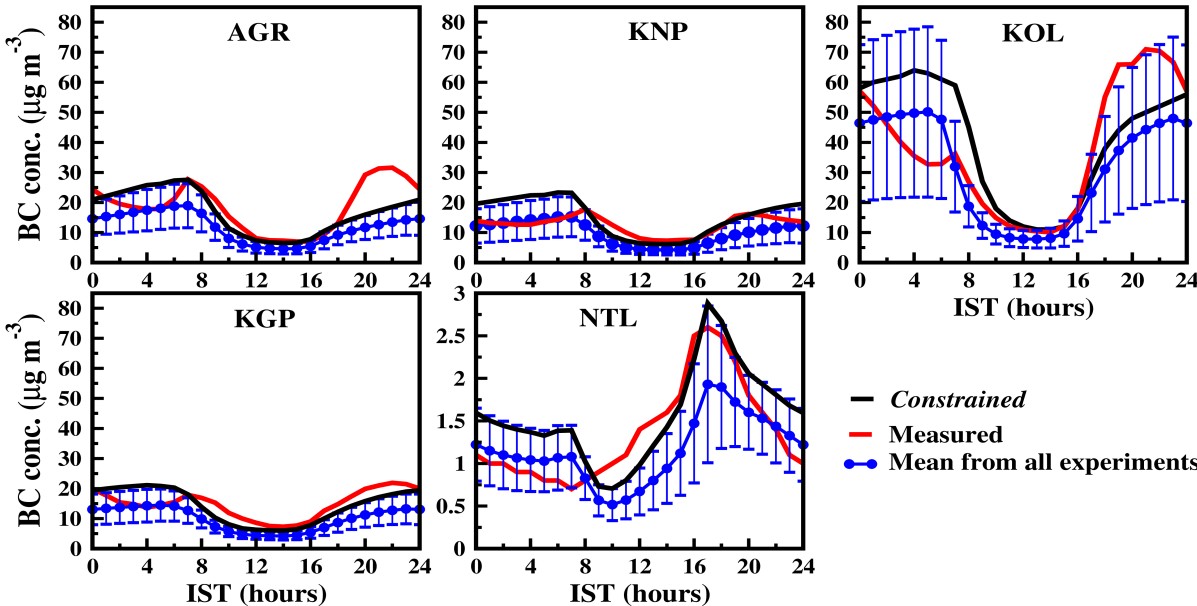

**Figure 4.** Hourly distribution of winter monthly mean of BC concentration ($\mu$g m$^{-3}$) at stations from $Constrained$ (black line). Note the y-axis is on a different scale for Nainital. The mean and the standard deviations ($1\sigma$) from the five simulations corresponding to the hourly winter monthly mean of BC concentration is also shown.

The mean and standard deviation of simulated BC concentration from five simulations at stations under study are provided in Table 3 (top) (also refer to refer to Figure 4). Analysis of multi-simulations indicates that the percentage deviation ($\delta$, refer to
415 equation 4) in simulated BC concentration (Table 3 (top)) attributed to emissions is, specifically, the lowest for the low polluted location (e.g., Nainital) and is, generally, within 40% for all other locations under study. The $\delta$ for the megacity is noted as being, typically, amplified (51%–56%) during the late evening to early morning hours (Table 3 (top), Figure 4) than that during daytime hours (35%–43%) compared to other locations under study. Thereby suggesting that under the similar meteorological condition and with the same aerosol processes in the model, the deviation in simulated BC concentration attributed to emissions
increases from daytime to late evening hours. Thus, indicating the increased emissions potentially amplify the accumulation of BC pollutants and the worsening of air quality over the megacities, specifically, during the late evening to morning hours than the daytime hours, raising concern for megacity commuters.

On comparing the temporal distribution of simulated BC concentration from the $Constrained$ with that of measured, it is seen that the pattern of simulated diurnal variability (shown for selected stations, refer to Figure 4) is consistent with
425 that of measured. The diurnal variability comprising of BC concentration being relatively higher, by a factor of 2 to 5 in $Constrained$, during the late evening to early morning hours (2000–0500 LT) than that during daytime hours (1000–1600 LT) at stations (except Nainital). Notably, this factor is equivalent to that obtained from $bottomup$ simulations and also to that from observations (Surendran et al., 2013; Pani and Verma, 2014; Ram and Sarin, 2010; Nair et al., 2012; Dumka et al., 2010; Lipi and Kumar, 2014). The diurnal variability in BC surface concentration is mainly associated with the atmospheric mixing depth

depending upon the stability characteristics of atmospheric layer linked with meteorology (Stull, 2012; Verma et al., 2013; Govardhan et al., 2015, 2019). It is worth noting that the specific feature observed in the temporal trend of BC concentration, comprising of peaked BC concentration during late afternoon hours (1500–1800 LT) at a high altitude location, Nainital, unlike the temporal trend at plain locations (e.g., Kolkata, Kharagpur). This specific feature conforms with measurements, and as inferred from available studies (Dumka et al., 2010; Stull, 2012) is attributed to the deepening of atmospheric mixing depth during the late afternoon hours which flushes out pollutants, including BC to the high altitude locations from the valley (Dumka et al., 2010; Stull, 2012). The distinct diurnal trend seen in BC surface concentration at Nainital in conjunction with the visualization of the animation (doi: http://doi.org/10.5446/48819, in supplementary material, also discussed later in the Section), thereby suggests the transport of BC pollution from the IGP towards the high altitude Himalayan station, Nainital. However, the BC emission strength at Nainital is relatively lower than the IGP (refer to Figure 1a). Therefore, consistent with observational studies, the simulated atmospheric BC concentration is noted to be the lowest at Nainital among stations under study.

The bias in the simulated hourly distribution of winter monthly mean of BC concentration (refer to Figure 4) with respect to observation is, however, noted to be larger by 40%–60% during mid-night to early morning hours (0000–0500 LT) than that during daytime hours. A larger bias is attributable to that the simulated aerosol processes in CHIMERE are influenced by the simulated diurnal meteorology from the WRF. It is to be noted that the WRF simulated hourly mean temperature is found being 45%–60% lower than the observed, specifically during 0000–0500 LT (as mentioned in Section 3.1), which has implications on the diurnal distribution of BC concentration. Besides, an enhanced accumulation of BC concentration during late evening hours (1800–2200 LT), specifically for megacity (e.g., Kolkata) and urban location (e.g., Agra), is noticed in the measurements compared to simulated values. Thereby indicating the requirement to improve the representation of the factors for hourly disaggregation of the total emission of pollutants in CHIMERE (Menut et al., 2012) during late evening hours (1800–2200 LT), specifically for megacity (e.g., Kolkata) and urban location (e.g., Agra). This improvement is suggested taking into account the enhanced traffic emissions from heavy-duty commercial vehicles (Ganguly et al., 2006; Bano et al., 2011; Kumar et al., 2020) at these locations during late evening h, hence needing a better representation of the factors accounting for this enhancement. The results of the diurnal BC distribution including improved representation of local emissions (specifically for megacity and urban locations) in CHIMERE forced by assimilated diurnal meteorological data will be presented in a future study.

We also provide an animation showing a representation of transboundary movement of BC pollutants over the IGP as a supplement (doi: http://doi.org/10.5446/48819, in supplementary material). This animation shows the hourly monthly mean of surface BC concentration to highlight the diurnal cycle, and its visualisation shows the diurnal evolution of the plume of BC surface concentration over the IGP. The BC surface plume is observed to be shrinking in magnitude during daytime hours (1000 LT–1600 LT) and swelling-up during evening till morning hours (1800 LT–0600 LT). It is visualised spreading towards the north (Himalayan side, Nainital) during afternoon hours (1200 LT–1800 LT), and towards the south (central India), and from the upper/northern IGP (e.g., Delhi) towards the lower/eastern IGP (e.g., Kolkata). The diurnal feature of surface BC plume distribution thereby appears to exhibit the pollution breathing pattern by the IGP region.

The megacity Delhi is surrounded by landmass on all sides and, as visualized from the animation, is influenced by the transport of pollutants from near-by regions (e.g., Punjab-Haryana) towards Delhi. In contrast, the megacity of Kolkata is a coastal location and the atmospheric BC concentrations are also affected by the prevailing land-sea breeze activity there (Verma et al., 2016). A relatively lower daytime mean BC concentration measured at Kolkata than Delhi (Figure 3f) is due to dilution of aerosol pollutants concentration with the relatively pristine maritime air mass (attributed to the prevailing low-intensity sea breeze during winter). In contrast to daytime mean, the higher all-day mean of BC surface concentration at Kolkata compared to Delhi (Figure 3g) is due to the outflow of BC pollutants from the upper/northern IGP towards eastern IGP at Kolkata (doi: http://doi.org/10.5446/48819, in supplementary material). The outflow is visualised to be comparatively stronger during the evening till early morning hours (1800 LT-0600 LT). Besides, the enhanced amplitude of BC concentration at Kolkata compared to Delhi during the late evening is also due to the increased accumulation of BC pollutants owing to the land-breeze activity during winter (Verma et al., 2016).

The correlation coefficient (r) between estimated and measured BC concentration for stations under study corresponding to each of the five simulations is also presented in Table 3 (bottom). A strong correlation is seen between model estimates and observations for both all-day and the daytime mean of BC concentration from each of the five experiments. The above analyses indicate that the temporal pattern, including the spatial trend (as discussed before) of BC distribution attributed to model processes (which govern the atmospheric residence time of BC, refer to Section 1) are simulated consistently well over the IGP, irrespective of the magnitude of the BC emission strength used in simulations.

**Table 3.** Top: Estimated mean and percentage deviation ($\delta$, refer to equation 4) of BC concentration from five simulations for locations under study; Bottom: Summary of statistical analysis comparing simulated BC concentration with measurements

| | Estimated mean and percentage deviation for all-day (daytime, late evening to early morning) mean of BC concentration from five simulations | | | | | | | | | |
|---|---|---|---|---|---|---|---|---|---|---|
| | Megacity | | Urban | | | | | Semi-urban | | Low polluted |
| Station | DEL | KOL | AGR | KNP | ALH | VRS | KGP | BBN | RAN | NTL |
| Mean ($\mu$g m$^{-3}$) | 38 (11, 55) | 33 (10, 48) | 12 (6, 15) | 10 (5, 12) | 12 (5, 16) | 13 (6, 18) | 10 (5, 13) | 8 (3.5, 11) | 6 (3, 9) | 1.2 (1, 1.5) |
| $\delta$ (%) | 41 (35, 51) | 54 (43, 56) | 33 (37, 37) | 40 (39, 45) | 40 (38, 41) | 37 (35, 37) | 30 (31, 36) | 35 (28, 38) | 31 (33, 31) | 25 (20, 18) |

| Summary of statistical analysis comparing simulated BC concentration with measurements for all-day (daytime) mean and performance evaluation | | | | | | |
|---|---|---|---|---|---|---|
| Experiment | r | NMB (%) | RMSE ($\mu$g m$^{-3}$) | Performance evaluation[a] | | |
| | | | | Best efficiency | Moderate efficiency | Low efficiency |
| *Smog* | 0.7 (0.9) | 38 (37) | 9 (3.5) | 4 (1) | 6 (7) | 0 (2) |
| *Edgar* | 0.8 (0.7) | 37 (57) | 9 (6) | 4 (1) | 2 (2) | 4 (7) |
| *Cmip* | 0.8 (0.9) | 52 (52) | 11 (5) | 0 (1) | 5 (4) | 5 (5) |
| *Pku* | 0.8 (0.9) | 45 (23) | 12 (2.5) | 5 (5) | 4 (5) | 1 (0) |
| *Constrained* | 0.9 (0.9) | 14 (17) | 3 (2) | 10 (10) | 0 (0) | 0 (0) |

[a]Number of stations out of total stations under study with the percentage bias in simulated estimates as $\leq\pm25\%$ (best efficiency), $>\pm25\%$ to $\pm50\%$ (moderate efficiency) and $>\pm50\%$ (low efficiency).

Further, to statistically evaluate the simulated BC concentration from each of the five simulations with respect to observations, we define the performance of the simulation considering the best, moderate, and poor efficiency based on their rela-

tive frequency to maintain the percentage bias in all-day (daytime) mean simulated BC concentration as about, respectively, $\leq\pm25\%$, $>\pm25\%$ to $\pm50\%$ and $>\pm50\%$ (refer to Table 3 (bottom)) corresponding to the observation data points under study. This consideration leads to identify $Constrained$ estimates delivering the best performance (percentage bias $\leq\pm25\%$) among all simulations for most of the times, i.e., for 100% (100%) of the total data points corresponding to measured value at stations under study. Estimates from $Pku$ exhibit the best performance for about 50% (50%) of the total stations. These from $Smog$ and $Edgar$ are for about 40% (10%) of the total stations under study. Estimates from $Smog$ and $Edgar$ are the most frequent respectively, corresponding to moderate and poor efficiency. Notably, unlike $Constrained$, the best efficiency is poorly frequent ($<$10%), specifically, for the daytime mean of BC concentration from $Smog$, $Edgar$, and $Cmip$; thereby, indicating the BC emission strength of respective emission database as input in the model are considerably low to simulate the BC distribution adequately over the IGP. A summary of statistical analysis with respect to Pearson correlation (r), NMB and RMSE accounted for the estimates of BC concentration from all simulations is also presented in Table 3 (bottom). The NMB (%) and RMSE ($\mu$g m$^{-3}$) values for the all-day (daytime) mean of BC concentration from $Constrained$ is about 14% (17%) and 3 (2) $\mu$g m$^{-3}$, which are the lowest among all of the simulations. The NMB from the $Constrained$ is noted being within the uncertainty limits reported in BC measurements (5%–20%).

### 3.3 Simulated wintertime BC-AOD with new BC emissions: Correlation analysis of variance

To evaluate the columnar distribution of wintertime BC aerosols over the IGP, the spatial distribution of the monthly mean of BC-AOD at 550 nm from simulations are presented in Figures 5a-e. The spatial pattern of BC-AOD distribution showing a large value over the IGP is consistent with the features of observed AOD from satellite retrievals (e.g., Verma et al., 2014). The value of BC-AOD distribution across the IGP from $Constrained$ (0.04–0.1) is found to agree well with that from a recent study (0.05–0.1) – based on a designed constrained aerosol simulation approach inferred being delivering a good agreement between model estimates and observations of atmospheric aerosol species (Kumar et al., 2018; Santra et al., 2019). The BC-AOD from $Constrained$ is also found to be matching consistently well (NMB: 11%) with absorption AOD (AAOD) from AERONET based observations at stations over the IGP (Kanpur, New Delhi-IMD, Gandhi College (25.87° N, 84.12° E), IIT Kharagpur extension at Kolkata) and BC-AOD estimated at Kolkata from the configured aerosol model using in-situ ground-based observations for Kolkata, (Verma et al., 2013)) (refer to Figure 5f). Estimated BC-AOD from simulations– $Pku$, $Smog$, $Cmip$ and $Edgar$, is lower in magnitude by, respectively, 15%–30%, 30%–50%, 40%–60% and 50%–70% than the $Constrained$ over most of the IGP. An overall comparison of the $Constrained$ and $bottomup$ estimates with measurements, thus, indicates the low magnitude and inconsistent spatial gradient across area types of the bottom-up BC emission flux as the primary reason for a large discrepancy in simulated BC concentration and BC-AOD from the $bottomup$.

The percentage BC-AOD fraction and BC mass fraction from $Constrained$ (Figures 5g-h), are estimated by taking the ratio of BC-AOD to total AOD and that of BC concentration to the total submicronic aerosol concentration, respectively. The total AOD and submicronic aerosol concentration required for estimating fractional distribution are obtained from a previous study (as mentioned above), based on the designed constrained aerosol simulation approach (Kumar et al., 2018). The BC-AOD fraction and BC mass fraction are about 10%–16% and 6%–10%, respectively, over most of the IGP. The estimated BC

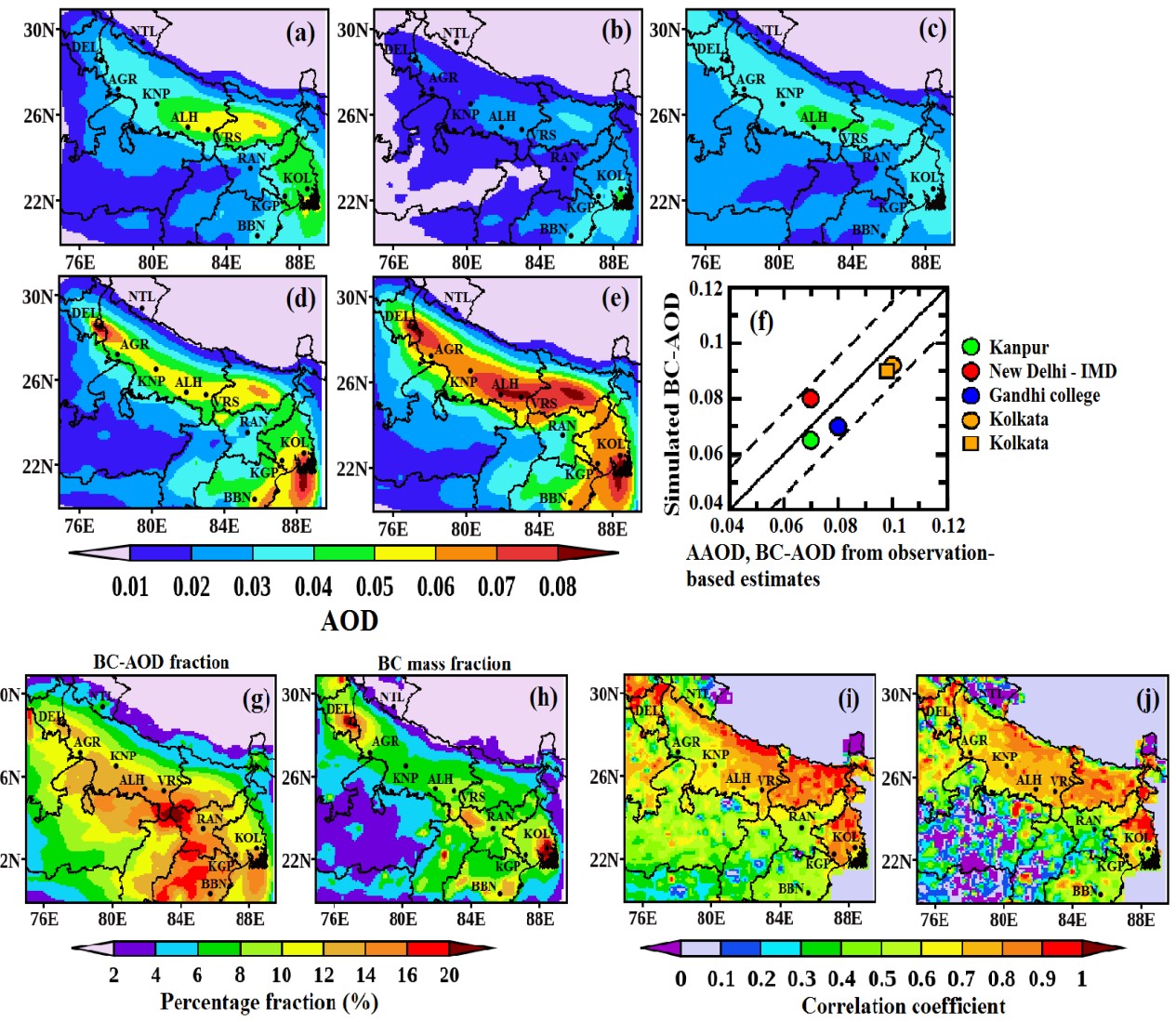

**Figure 5.** (a–e) Spatial distribution of simulated winter monthly mean of BC-AOD at 550 nm over IGP from (a) *Smog*, (b) *Edgar*, (c) *Cmip*, (d) *Pku*, and (e) *Constrained*; (f) comparison of simulated BC-AOD from *Constrained* with the AAOD –represented as circles (from AERONET based observations at stations (Kanpur, New Delhi-IMD, Gandhi College, IIT Kharagpur extension at Kolkata)), and with BC-AOD – represented as a square estimated using in-situ ground-based observations at Kolkata; the dashed lines correspond to the value within ±25% of the 1:1 comparison shown as solid line; (g–h) spatial distribution of (g) BC-AOD fraction (%), and (h) BC mass fraction (%) from *Constrained*; (i–j) correlation between variance in BC emission flux from five BC emission databases and in estimated (i) BC-AOD, (j) BC concentration from five simulations.

mass fraction in the present study is also seen to be in corroboration with values reported from wintertime measurements over the Indian region, e.g., noted as being 12% (wintertime average) of the total submicronic aerosol concentration over Kolkata, 4%–15% of the total aerosol concentration over Delhi and Kanpur, 3%–7% of $PM_{2.5}$ over Varanasi and Anantpur, including that over Kaashidhoo climate observatory in Maldives (Verma et al., 2013; Kumar et al., 2017; Tripathi et al., 2005a; Ganguly et al., 2006; Reddy et al., 2012; Satheesh et al., 1999). The location of hotspots for BC mass fraction (value $> 16\%$), BC-AOD fraction (12%–16%), including that for BC-AOD (value $> 0.08$) is seen to overlap with that identified for BC surface concentration (Figure 3e). It is also seen that the percentage fraction of BC-AOD, in general, is about twice larger than the BC mass fraction, indicating that even a low BC concentration in aerosol mass has the potential to contribute significantly to attenuation of solar radiation and thereby influence the regional radiation balance (which is examined in the next section).

To gain insight into the degree of association of the simulated BC burden with the BC emission strength, we utilise the five simulations to evaluate the correlation coefficient between the variation in emission strength and that in simulated BC-AOD (Figure 5i) or simulated BC concentration (Figure 5j). A strong correlation (correlation coefficient: $>0.7$) is seen between the variance in BC emission and simulated BC mass concentration or BC-AOD, e.g., as observed over most of the IGP region (including megacity Kolkata). The strong correlation is indicative that the change in BC emission flux primarily governs change in the simulated BC mass concentration and BC-AOD. On the other hand, a moderate correlation (correlation coefficient: 0.5–0.6), e.g., as observed over parts of lower-mideastern IGP (patch 'P5', refer to Figure 3e), over parts of northern IGP (patch 'P1') including in and around megacity Delhi for both BC surface concentration and BC-AOD, and over some parts of eastern IGP (patch 'P4'), including the area around megacity Kolkata for BC surface concentration. The moderate correlation suggests that over these parts, besides the change in BC emission strength, transport of BC aerosols as governed by model processes also have a profound impact on influencing the simulated BC burden. In other words, a large BC burden over the megacity Delhi region is profoundly impacted due to the transport of BC aerosols, besides the BC emission strength. A weak correlation (correlation coefficient: $< 0.5$), e.g., as observed over Nainital and Ranchi stations and the surrounding area, suggests the transport of BC aerosols compared to the BC emission strength primarily influences the BC burden. It is also noted that over the central region (bounded between 76°E–80°E and 20°N–26°N), correlation for BC-AOD is moderate, but, however, is still stronger than BC concentration. Thereby indicating the potential influence to BC-AOD over the region from high rise BC emissions (corroborated by prevalence of open biomass burning emissions, Venkataraman et al. (2006)) and the elevated transport of BC aerosols as also inferred in a previous study (Verma et al., 2008).

## 3.4 Wintertime direct radiative perturbations due to BC aerosols: comparison with atmosphere eliminating BC

Further, the wintertime SW direct radiative perturbation due to BC aerosols over the IGP (Figures 6a-c) is evaluated corresponding to the layers of atmosphere (SUR, ATM, and TOA, refer to Section 2.3). We also compare the direct radiative perturbation due to BC with that estimated considering the atmosphere eliminating or without BC aerosols (Figures 6d-f) to evaluate the magnitude of direct radiative perturbation in the presence of BC aerosols. The positive value of radiative effect signifies warming due to BC aerosols and is vice versa for the negative value of the radiative effect. There is a reduction in the wintertime radiative flux due to BC at the SUR by $-20$ to $-40$ W m$^{-2}$ (Figure 6a). The radiative warming (Figure 6c)

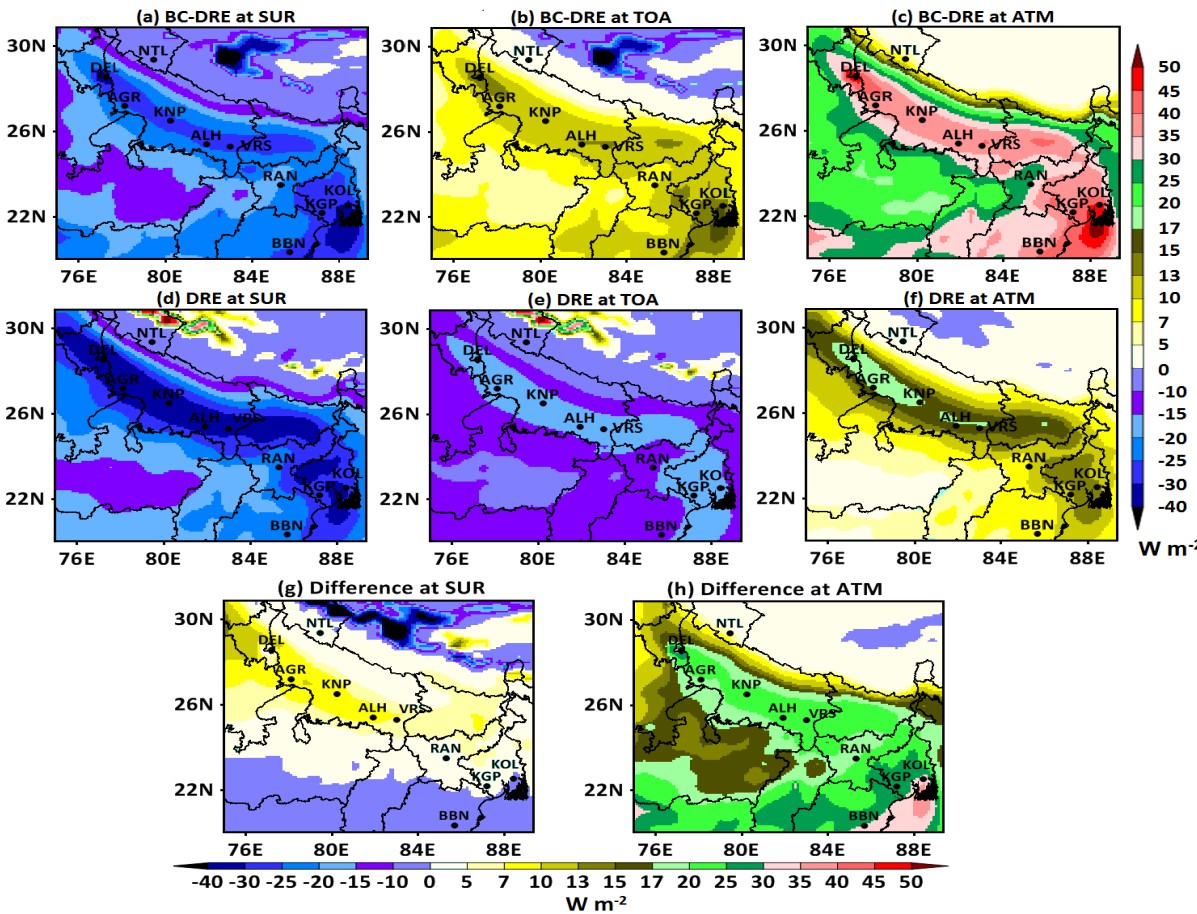

**Figure 6.** (a–c) Spatial distribution of wintertime direct radiative perturbation due to BC aerosols from $Constrained$ at (a) SUR, (b) TOA, and (c) ATM; (d–f) same as (a–c) but with atmosphere eliminating BC; (g–h) difference in radiative perturbation due to BC and atmosphere eliminating BC at (g) SUR, and (h) ATM.

due to BC aerosols at the ATM ($+30$ to $+50$ W m$^{-2}$) is estimated to be about 50%–70% larger than the cooling due to BC at the SUR. The magnitude of SUR cooling effect as noted due to BC aerosols is, however, found to be 10%–20% lower than that estimated considering the atmosphere eliminating BC aerosols (Figures 6d and 6g). Moreover, the magnitude of ATM radiative warming due to BC is seen to be larger by 2–3 times compared to the atmosphere without BC aerosols (Figures 6f and 6h). The radiative effect at the TOA due to BC aerosols (Figure 6b) is positive and, thereby, indicates a net radiative warming effect ($+10$ to $+17$ W m$^{-2}$) over the IGP during winter. In contrast, a cooling effect at TOA ($-10$ to $-20$ W m$^{-2}$) is exhibited considering the atmosphere without BC aerosols (Figure 6e). It is also seen that the patch with the most substantial value ($>15$ W m$^{-2}$) of the net radiative forcing due to BC is observed in and around megacities and is extended to the eastern coast. A comparison of the radiative effect due to BC from $Constrained$ estimates with that from $Smog$ estimates shows that bottom-up BC emissions (e.g., Smog-India) lead to a relatively lower wintertime radiative warming at ATM and TOA by 30%–50% than the constrained emissions over most of the IGP and that by more than 80% over northern IGP (in and around

Delhi). The comparison between the *bottomup* and the *Constrained* estimates, thus, indicates the potential underestimation of wintertime direct radiative perturbation due to BC aerosols over the IGP attributable to the low BC emission strength in the bottom-up BC emission database.

The uncertainty in estimated wintertime direct radiative perturbations in the present study is inferred to be within 40%. This estimation is based on taking into account NMB in simulated BC concentration (as presented in Section 3.2) and the model variability (33%) in estimated DRF of BC based on the evaluation of twenty global aerosol models (Schulz et al., 2006).

## 4  Conclusion

In the present study, wintertime direct radiative perturbation due to black carbon (BC) aerosols were examined over the Indo-Gangetic plain (IGP) evaluating the efficacy of the fine grid resolved ($0.1° \times 0.1°$) BC aerosol transport in a chemical transport model (CHIMERE); offline-coupled with WRF regional meteorological model. The efficacy of CHIMERE to simulate the observed BC surface concentration was assessed implementing the new BC emission inventories and through a detailed validation and statistical analysis of simulated BC concentration with respect to ground-based measurements at stations over the IGP. The five BC transport simulations, *Constrained* and *bottomup* (*Smog*, *Cmip*, *Edgar*, and *Pku*) were performed implementing BC emission data, respectively, from India-based 'constrained' and bottom-up 'Smog-India' and that from the three global bottom-up 'CMIP6', 'EDGAR', and 'PKU' extracted over the Indian region.

The WRF simulated winter monthly mean of the meteorological parameters resembled well (bias $<\pm25\%$) the measured counterparts. However, the diurnal distribution of meteorological parameters exhibited a considerable discrepancy, specifically during mid-night to early morning hours (0000 LT to 0500 LT), having implications on the diurnal distribution of BC concentration. In this regard, a better temporally resolved meteorological boundary condition in WRF, aided with data assimilation at a fine temporal resolution (e.g., 1-hourly) using observations for India-based stations, needs to be assessed in a future study.

A strong association of the winter monthly mean BC concentration between modelled and measured values for stations under study corresponding to each of the five simulations was noticed. The efficacy to simulate the magnitude of observed wintertime BC distribution was found to be moderate to poor for *bottomup*. The *Constrained* estimated large BC pollution over the IGP comprising of wintertime all-day monthly mean BC surface concentration (BC-AOD) as 14–25 $\mu$g m$^{-3}$ (0.04–0.08) resembled well the observed counterparts. These estimates were noted with the lowest percentage bias ( $\leq\pm25\%$) among five simulations for each of the stations and area types under study. The low magnitude and inconsistent spatial gradient across area types of the bottom-up BC emission flux was found as the primary reason for a large discrepancy in simulated BC concentration and BC-AOD from the *bottomup*.

The BC-AOD fraction (10%–16%) from the *Constrained* was noted to be about twice larger than the BC mass fraction (6%–10%) over most of the IGP region. Five hotspots comprising of large BC load (surface concentration $>16$ $\mu$g m$^{-3}$ from *Constrained*), were identified in and around megacities (Delhi and Kolkata) and surrounding semi-urban area, urban spots over central and mideastern IGP (Prayagraj/ Allahabad–Varanasi, Patna), and including the rural spot over the lower mideastern-IGP (Palamu).

Analysis of multi-simulations of BC transport in CHIMERE, indicated the increased emissions in the megacities potentially amplify the accumulation of BC pollutants, specifically, during the late evening to morning hours, raising concern for megacity commuters. The correlation between the variance in emissions and simulated BC mass concentration and BC-AOD from the five simulations manifested the sensitivity of simulated BC concentration and BC-AOD primarily to the change in BC emission strength over most of the IGP (including megacity Kolkata). But that also to the transport of BC aerosols as governed by model processes over megacity Delhi and the area around megacities of Delhi and Kolkata.

The transboundary movement of the wintertime BC plume of the IGP was visualized to be spreading towards the north (Himalayan side) during afternoon hours (1200 LT–1800 LT). And towards the south (central India) and from the upper/northern IGP (e.g., Delhi) towards the lower/eastern IGP (e.g., Kolkata) during evening till morning hours (1800 LT–0600 LT).

Analysis of direct radiative perturbations due to BC aerosols showed that wintertime BC aerosol over the IGP enhances the atmospheric warming by 2–3 times more, and, reduces the surface cooling by 10%–20% lesser than considering the atmosphere eliminating BC aerosols. The BC induced net warming effect at the top of the atmosphere (TOA) from the $Constrained$ was estimated as 10–17 W m$^{-2}$ over most of the IGP, in contrast to a net cooling at the TOA considering the atmosphere without BC. The radiative perturbation was spotted being spatially the largest in and around megacities (Kolkata and Delhi) and extended to the eastern coast. These were assessed to be about 30%–50% lower from the $bottomup$ than the $Constrained$ over most of the IGP.

The present study showed that an adequate BC emission strength and a meteorological forcing in a state-of-the-art chemical transport model at a fine grid resolution led to successfully simulate the wintertime BC distribution (surface concentration and BC-AOD) over the IGP, unlike previous studies (refer to Section 1). We believe this distribution provided a reasonably more accurate representation of the simulated wintertime direct radiative perturbations due to BC aerosols, including identifying the BC hotspots over the IGP. The wintertime radiative perturbation due to BC aerosols simulated in the present study is further utilized to evaluate the potential response on temperature, air quality, and regional climate over the IGP; the outcome from these evaluations will be presented in a future study. The present study is also further extended to evaluate the inter-seasonal BC distribution and associated radiative impacts over the Indian subcontinent with their implications on the southwest monsoon rainfall.

*Supplementary material*. The animation is available at https://av.tib.eu/media/48819 (doi: http://doi.org/10.5446/48819).

*Data availability*. The data in this study are available from the corresponding author upon request (shubha@iitkgp.ac.in).

*Author contributions*. SG conducted the BC transport simulations and radiative transfer simulations, evaluation, and validation of the model estimates, including the statistical analyses, and participated with SV in synthesizing and analyzing the results. SV planned and coordinated the study. SG and SV wrote the paper. JK and LM contributed to the writing and analysis of results. LM also advised for the technicality of the CHIMERE model configuration.

*Competing interests*. The authors declare that they have no conflict of interest.

*Acknowledgments*. This work was supported through a grant received for the project National Carbonaceous Aerosol Programme–Carbonaceous Aerosol Emissions, Source Apportionment and Climate impacts (NCAP–COALESCE) from the Ministry of Environment, Forest, and Climate Change (14/10/2014-CC (Vo. II)), Govt. of India at the Indian Institute of Technology, Kharagpur. Simulations were performed in a high-performance computing cluster developed at IIT-KGP supported through the NCAP-COALESCE. Contributions of Mr. Rhitamvar Ray, Project Engineer at IIT-KGP supported by NCAP-COALESCE project, are duly acknowledged towards maintaining the computing cluster, handling simulations, data extraction, and preparing the BC animation.

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
