# Peer review of "Wintertime direct radiative effects due to black carbon (BC) over Indo-Gangetic Plain as modelled with new BC emission inventories in CHIMERE"

_Atmospheric Chemistry and Physics, 2020_

## Referee Comment (RC1) · Anonymous Referee #2 · 30 Oct 2020

Wintertime radiative effects of black carbon (BC) over Indo-Gangetic Plain as modelled with new BC emission inventories in CHIMERE This manuscript discussed the radiative perturbation due to black carbon (BC) with modelled BC distribution in a high resolution  $(0.1 \times 0.1)$  chemical transport model, CHIMERE over Indo Gangetic plain (IGP) during winter period when pollution level load in high amounts. They performed multiple BC transport simulations with CHIMERE and put into practice with new BC emission inventories, which included the recently estimated India-based constrained BC emissions and the latest bottom-up BC emissions (India-based: Speciated Multi-pOllutant Generator (Smog-India), and global: Coupled Model Inter comparison Project phase 6 (CMIP6), Emission Database for Global Atmospheric Research-V4 (EDGAR-V4) and

Peking University BC Inventory (PKU)). Aerosol optical depth due to BC (BC-AOD) and its fractional contribution to total AOD are also highlighted in the study. The paper is well written and straightforward. Therefore, I recommend the manuscript for publication after some changes have been taken into account. 1. In the Abstract more quantitative information should be included 2. It would be good, if authors can include more about emission estimation of BC at global and India level with uncertainty in the introduction section which will be useful for the readers. Authors may follow these references Bounding the role of black carbon in the climate system: a scientific assessment. J. Geophys. Res., 118 (2013), pp. 5380-5552, 10.1002/jgrd.50171. Residential biofuels in south Asia: carbonaceous aerosol emissions and climate impacts Science, 307 (2005), pp. 1454-1456 Line 30: Authors may relate this sentence with CO2 in terms of warming. 3. Line 290-293: Accumulation is the only cause of leading values or night time heavy transport vehicles emissions may be another reason. Please discuss it. 4. Line 315-319; what is cause for the lower values of BC at high altitude. Is there no impact of transport of air masses? Please explain it in detail. 5. Figure 3(f-g); Delhi and Kolkata are megacities but BC measured high in Delhi during the day time and low in Kolkata for the same period. Why? Please elaborate. 6. The captions are too long of most of the figures. If possible, please short it to better readability. 7. At few places' sentences are bit long and complex so these long sentences should be broken into small sentences which will be helpful for readers 8. Though authors have discussed their findings but still I feel more discussion about BC concentrations over IGP region. 9. The conclusion section should be short and crispy for better readability.

---

## Author Comment (AC1) · 21 Dec 2020

Reply to query, point 4 by Referee 2:

Dear Referee: Thank you for your comments. We are incorporating your suggestions in the next version of the manuscript. Here we discuss your query, point 4. Please let us know if any further suggestions are there.

4. Line 315-319; what is cause for the lower values of BC at high altitude. Is there no impact of transport of air masses?

Response: Thank you for an interesting question. In our analyses presented for vari-
ous stations, Nainital is a high altitude station and is classified as a low-polluted station. As seen from Figure 1, the BC emission strength at Nainital is relatively lower than the IGP. Hence, as expected and consistent with observational studies, the simulated atmospheric BC concentration is found to be the lowest at Nainital among stations under investigation. The spatial pattern (refer to Figures 3f-g) of BC surface concentration while exhibiting the lowest value at high altitude and low-polluted location (e.g., Nainital), and the moderately high values at semi-urban stations (e.g., Kharagpur and Ranchi) is seen to reach the maximum at megacities (Kolkata and Delhi). The simulated spatial pattern is consistent with observation.

Yes, Nainital is influenced by transport of BC pollution from the IGP. We request you to kindly watch the animation showing a representation of transport of BC concentration over the IGP as a supplement (please see BC-animation-1 in supplementary material). This animation shows the hourly monthly mean of surface BC concentration to highlight the diurnal cycle and its visualisation shows the diurnal evolution of the BC plume over the IGP. The BC surface plume is observed to be shrinking during daytime hours (1000 LT–1600 LT) and swelling-up during late evening till morning hours (1800 LT–0600 LT) when it is visualised spreading towards the south (central India) and north (Himalayan side) and also from the upper/northern IGP towards the lower/eastern IGP. The diurnal feature of surface BC plume distribution thereby appears exhibiting the pollution breathing pattern by the IGP region.

As visualised from the animation and our analysis presented in the manuscript (lines 362-368), it is seen that there is transport of BC pollution from the IGP towards Nainital, though, atmospheric dispersion is inhibited by the Himalayan mountains. As discussed in Section 3.1, there is a confinement of pollution near the surface within the shallow boundary layer height in winter due to low vertical mixing and weak dispersion of atmospheric pollutants, thereby, stagnant weather under the prevailing meteorological conditions, viz. low temperature and weak wind speed, the downdraft of the air mass, and a narrow PBLH (refer to Section 3.1). Besides, the Himalayan mountains northward, further, inhibits the dispersion of aerosol pollutants and favours their confinement over the IGP.

The diurnal variability in BC surface concentration presented in Figure 4 further confirms the influence of transport of BC pollution at Nainital. It is worth noting that the specific feature observed in the temporal trend of BC concentration (refer to Section 3.2, Lines 340-345), comprising of peaked BC concentration during late afternoon hours (1500–1800 LT) at high altitude location, Nainital, unlike the temporal trend observed at plain locations (e.g., Kolkata, Kharagpur), conforms with measurements. This specific feature, as inferred from available studies is attributed to the deepening of atmospheric mixing depth during the late afternoon hours which flushes out pollutants, including BC to the high altitude locations from the valley.

---

## Author Comment (AC2) · 22 Dec 2020

Reply to Referee #1 on their previous comment and request for further comments if any

Dear Referee:

Thank you for your valuable comments and suggestions for the manuscript previously. Specific changes were made in response to the comments (please see below) and were posted in Author's response file uploaded on September 24, 2020. We also uploaded the revised manuscript with suggestions implemented on September 24, 2020.

Please let us know of any of your comments further.

Referee #1: It is interesting to learn that bottom up emissions associated results near megacities are found to be 30-50% lower as compared to constrained. I am not really sure how one can generalise this. Nevertheless, I do not have any concerned at this stage accepting this paper for discussion phase of the ACP. Figure quality is very poor and difficult to visualise due to choice of colours.

Response:

Thank you for the suggestion. As per Reviewer's suggestion, figures have been improved, specifically in their color scales, to represent better the spatial features. It is also to be noted that the degree of comparison is examined with the estimated BC concentration from five simulations subjected to the same aerosol physical and chemical processes with CHIMERE. The mentioned "30-50%" correspond to BC radiative effect values. Please note that Figure 6 for radiative effects is provided only for Constrained simulation (using the BC distribution, which is found to represent well the observed values).

The BC radiative effects from the bottomup simulations (using bottom-up BC emissions, e.g., Smog-India) is found to be lower than that from Constrained simulation (using constrained BC emissions). We have evaluated the performance of the new BC emissions (bottom-up and constrained), with a state-of-the-art chemical transport model, towards their adequacy to represent the BC distribution and thereby, the climatic impacts over the IGP in the Indian subcontinent. ACPD

---

## Author Comment (AC3) · 27 Dec 2020

Reply to Referee #3 on their previous comment

Dear Referee:

Thank you for your valuable comments and suggestions for the manuscript previously. Specific changes were made in response to the comments (please see below) and were posted in Author's response file uploaded on September 24, 2020. We also uploaded the revised manuscript with suggestions implemented on September 24, 2020. Please let us know of any of your comments further.

[Figure]

Referee #3:

The submitted manuscript examines the sensitivity of radiative effects of black carbon (BC) over the IGP of India using offline model CHIMERE and five different emission inventories. I noticed that there is no discussion on the BC emission inventories and why they differ in their estimate over India. I would therefore suggest to at least adding a better description of emission inventories in the stage to the discussion document. What is the motivation of selecting wintertime is not clearly coming in the introduction. Over all MS is ok and can be sent out for discussion and review.

Response: Thank you for the suggestions. As per Reviewer's suggestion, we have further provided more details on emission inventory description. Please also see Section 2.1.3 and refer to Table 1.

A suggested improvement in Introduction for wintertime motivation is also included. In addition to the above, we have also now introduced a more clarity in usage of "offline" term for the model description (refer to Section 2.1). The CHIMERE is a chemical transport model (CTM). The CTM needs to be forced by meteorology. The CHIMERE (model version 2014b) configuration in the present study is forced externally by Weather Research and Forecasting (WRF-V3.7) model as a meteorological driver in offline mode, meaning that the meteorology is pre-calculated with WRF then read in CHIMERE. In case of our study, this configuration has an interest since we are performing emission scenarios. Having calculated the meteorology one time, we are sure that the differences between the simulations are due and only due to emission scenarios and not to possibly chaotic retroactions due to an online coupling between meteorology and aerosols.

---

## Author Comment (AC4) · 27 Dec 2020

Reply to Referee #4 on their previous comment

Dear Referee:

Thank you for your valuable comments and suggestions for the manuscript previously. Specific changes were made in response to the comments (please see below) and were posted in Author's response file uploaded on September 24, 2020. We also uploaded the revised manuscript with suggestions implemented on September 24, 2020. Please let us know of any of your comments further.

[Figure]

Referee #4: Authors estimated radiative impact of black carbon (BC) aerosols over IndoGangetic Plain (IGP) using high resolution (0.1°×0.1°) chemical transport model (CHIMERE). Initially authors assessed the new BC emission inventories (Constrained, bottomup - Smog , Cmip , Edgar , and Pku) and through a detailed validation and statistical analysis of simulated BC concentration with respect to ground-based measurements at several stations over the IGP (both spatial and temporal variations). The draft manuscript is generally well organized and written. Considering these, I strongly recommend this paper for publication in ACPD.

General comments: 1. The definition of Bias, NMB, RMSE are well known to the community and authors need not to present them here. Response: We understand Reviewer's view. But still for clarity on calculations, we keep the formulations in the present version of the manuscript.

2. Lines # 255 and # 260 need appropriate references to support the attributes. Response: This is done.

3. From the several publications, I noticed that BC mass fraction reported in this study over IGP is similar to the other Indian sites and also Kaashidhoo (Maldives), and Bay of Bengal. Can authors comments on this?

Response: Thank you for the suggestion. The above information is now incorporated. These analyses thus indicate that the BC mass fraction and BC-AOD fraction maintains relatively uniform values or a lower spatial variability compared to BC mass concentration and BC-AOD.

4. Author may change the level of x-axis of Fig 4 as 0, 4, 8. . . 24 instead 0, 5, 15, 20.

Response: Thank you for the suggestion. This is done.

---

## Referee Comment (RC2) · Anonymous Referee #5 · 30 Dec 2020

This work estimated radiative impact of black carbon (BC) aerosols over one of the most polluted region in the world, IndoGangetic Plain (IGP) using a high resolution (0.1°×0.1°) chemical transport model simulation. Authors assessed the five BC emission inventories (Constrained, bottomup - Smog , Cmip , Edgar , and PKU) and through a detailed statistical analysis of simulated BC concentration with respect to measurements, and constraint the large uncertainty of direct radiative forcing (RF) of BC. My specific observations are following, hope these could help improve the manuscript and make it better suitable for publishing in ACP.

1) Since it is an offline modeling study, only direct RF can be assessed but not indirect

RF. This point deserves a further stress in the title and abstract. Also, I think some discussion regarding the potential influence (direct/indirect adjustment) of BC on cloud and cloud associated RF is worth to be commented in the manuscript.

2) Details of each BC Emission inventory is missing. More details are appreciated here. Such as what emission sectors are considered, how seasonal variation are considered (only Dec. is simulated) and etc.

3) Could you explain what do you mean by "constrained emission inventory"? Do you mean top-down based on satellite observation? More details about this is also required in the method.

4) Maybe I overlooked, but, it seems authors forgot to describe the AERNET AAOD dataset in the Method section.

5) line-159: SO2 is not aerosol

6) As best as I know, the lastest EDGAR emission is with 0.1*0.1 deg. resolution. Please double check and correct it accordingly in Table 1.

7) As best as I know, WRF model is km-based, not deg.-based, so how could possible to perform WRF simulation in 0.1*0.1 deg. resolution?

8) I think more validation of model results would be needed. I not sure if it is appropriate to validate the model (2015 simulation) with observations in about 10 years ago (2004. . .2011, in Table 2). If most of observations are available in 2004, why not perform simulation in 2004, what is your specific reason to simulate 2015? And, only validate the surface meteo. may not be enough, some validation of vertical profiles would be appreciated, since radiative transfer is a vertical path.

9) Why December is chosen for the simulation, any specific reason? I see in the introduction, authors mention that BC concentration is high in winter IGP, but this high concentration is with respect to surface concentration, however, this paper focuses on radiative forcing which is a vertical column transfer path. High surface pollutants

concentrations in winter IGP could be solely due to shallow PBL in winter compared to summer (Chen et al., 2020), but not necessarily a high column loading which is more relevant to RF. Some comments of this could help strengthen the discussion.

Reference Chen, Y., Wild, O., Conibear, L., Ran, L., He, J., Wang, L., and Wang, Y.: Local characteristics of and exposure to fine particulate matter (PM2.5) in four indian megacities, Atmospheric Environment: X, 5, 100052, https://doi.org/10.1016/j.aeaoa.2019.100052, 2020.

---

## Author Response (AR1)

**Response to Reviews of manuscript entitled "Wintertime radiative effects of black carbon (BC) over Indo-Gangetic Plain as modelled with new BC emission inventories in CHIMERE (ACP-2020-511)"**

We thank the Editor and Reviewers for their valuable comments, suggestions, and corrections. Specific changes made in response to the comments are described below.

**Response to Referees' comments**

**Referee 2:**

**General comment:** Wintertime radiative effects of black carbon (BC) over Indo-Gangetic Plain as modelled with new BC emission inventories in CHIMERE This manuscript discussed the radiative perturbation due to black carbon (BC) with modelled BC distribution in a high resolution ($0.1° \times 0.1°$) chemical transport model, CHIMERE over Indo Gangetic plain (IGP) during winter period when pollution level load in high amounts. They performed multiple BC transport simulations with CHIMERE and put into practice with new BC emission inventories, which included the recently estimated India-based constrained BC emissions and the latest bottom-up BC emissions (India-based: Speciated Multi-pOllutant Generator (Smog-India), and global: Coupled Model Inter comparison Project phase 6 (CMIP6), Emission Database for Global Atmospheric Research-V4 (EDGAR-V4) and Peking University BC Inventory (PKU)). Aerosol optical depth due to BC (BC-AOD) and its fractional contribution to total AOD are also highlighted in the study. The paper is well written and straightforward. Therefore, I recommend the manuscript for publication after some changes have been taken into account.

Response: Thank you for sparing time reading our manuscript. The comments and suggestions have been helpful to improve our manuscript. Please find the replies to specific comments herewith:

1. In the Abstract more quantitative information should be included

Response: This is done. We have checked the Abstract for more quantitative information as and where necessary and revised it accordingly.

2. It would be good, if authors can include more about emission estimation of BC at global and India level with uncertainty in the introduction section which will be useful for the readers. Authors may follow these references Bounding the role of black carbon in the climate system: a scientific assessment. J. Geophys. Res., 118 (2013), pp. 5380-5552, 10.1002/jgrd.50171. Residential biofuels in south Asia: carbonaceous aerosol emissions and climate impacts Science, 307 (2005), pp. 1454-1456.

Response: This is done. Please see Section 1, lines 80–86 in the revised manuscript. The new and revised lines included are also given below:

Section 1, Lines 80–86:
"The uncertainty in bottom-up BC emission inventory has been inferred about greater than 200% over India and Asia (Bond et al., 2004; Streets et al., 2003; Venkataraman et al., 2005; Lu et al., 2011), compared to that about 40% in recently estimated constrained BC emission over India (Verma et al., 2017). Therefore, in the above context, it is necessary to evaluate the divergence in BC emission flux from state-of-the-art bottom-up BC emission inventories and constrained BC emission towards examining the improvement of BC emission source strength required over the Indian region. Furthermore, it is also required to assess the efficacy of simulating the BC burden in a state-of-the-art chemical transport model under the different emission scenarios (e.g., bottom-up and constrained)."

Line 30: Authors may relate this sentence with $CO_2$ in terms of warming.

Response: This is done. Please see lines 32–34 in the revised manuscript. The revised lines are also given below:

Section 1 Lines 32–34:
"Among aerosol constituents, BC aerosols are considered the strongest absorber of visible solar radiation and, thereby, a prominent contributor to tropospheric warming as for the greenhouse gases carbon dioxide and methane (Ramanathan and Carmichael, 2008; Gustafsson and Ramanathan, 2016; Masson-Delmotte et al., 2018)".

3. Line 290-293: Accumulation is the only cause of leading values or night time heavy transport vehicles emissions may be another reason. Please discuss it.

Response: We have revised the sentences of this part (Section 3.2) to include more information about heavy transport vehicle emissions. The revised text is:

Section 3.2, Lines 439–448:
"Besides, an enhanced accumulation of BC concentration during late evening hours (1800–2200 LT), specifically for megacity (e.g., Kolkata) and urban location (e.g., Agra), is noticed in the measurements compared to simulated values. Thereby indicating the requirement to improve the representation of the factors for hourly disaggregation of the total emission of pollutants in CHIMERE (Menut et al., 2012) during late evening hours (1800–2200 LT), specifically for megacity (e.g., Kolkata) and urban location (e.g., Agra). This improvement is suggested taking into account the enhanced traffic emissions from heavy-duty commercial vehicles (Ganguly et al., 2006; Bano et al., 2011; Kumar et al., 2020) at these locations during late evening hours, hence needing a better representation of the factors accounting for this enhancement."

4. Line 315-319; what is cause for the lower values of BC at high altitude. Is there no impact of transport of air masses? Please explain it in detail.

Response: This is done. We have now included more explanation in Section 3.2, lines 429–433. Please also refer to animation showing a representation of transport of BC concentration over the IGP as a supplement to the manuscript. The revised lines in the manuscript are given below:

"The distinct diurnal trend seen in BC surface concentration at Nainital in conjunction with the visualization of the animation (doi: http://doi.org/10.5446/48819, in supplementary material, also discussed later in the Section), thereby suggests the transport of BC pollution from the IGP towards the high altitude Himalayan station, Nainital. The BC emission strength at Nainital is relatively lower than the IGP (refer to Figure 1a). Therefore, consistent with observational studies, the simulated atmospheric BC concentration is noted to be the lowest at Nainital among stations under study."

5. Figure 3(f–g); Delhi and Kolkata are megacities but BC measured high in Delhi during the day time and low in Kolkata for the same period. Why? Please elaborate.

Response: This is done. We have included the explanations in our revised version of the manuscript. Please see Section 3.2, lines 396–402, and lines 457–467. The lines included in manuscript are also given below:

Section 3.2, Lines 396–402:
"The simulated spatial pattern is consistent with observations (Figures 3f–g). The features of BC distribution for the specific area types are represented well by the simulated BC distribution. The spatial feature also indicates that while the BC emission strength (refer to Figure 1a) of the two megacities, Delhi and Kolkata, is nearly equivalent. But the wintertime all-day mean value of BC concentration (Figure 3g) at Delhi is lower than Kolkata, and is vice versa for the daytime mean value (Figure 3f). We discuss these features in context with the transport of BC aerosols over the IGP based on the visualization of an animation (doi: http://doi.org/10.5446/48819, in supplementary material) later in the Section."

Section 3.2, Lines 457–467:
"The megacity Delhi is surrounded by landmass on all sides and, as visualized from the animation, is influenced by the transport of pollutants from near-by regions, northwards to Delhi (e.g., Punjab-Haryana), towards Delhi (please see animation). In contrast, the megacity of Kolkata is a coastal location and the atmospheric BC concentrations are also affected by the prevailing land-sea breeze activity there (Verma et al., 2016). A relatively lower daytime mean BC concentration measured at Kolkata than Delhi (Figure 3f) is due to dilution of aerosol pollutants concentration with the relatively pristine maritime air mass (attributed to the prevailing low-intensity sea breeze during winter). In contrast to daytime mean, the higher all-day mean of BC surface concentration at Kolkata compared to Delhi (Figure 3g) is due to the outflow of BC pollutants from IGP towards eastern India at Kolkata (doi: http://doi.org/10.5446/48819, in supplementary material). The outflow is visualised to be comparatively stronger during the late evening till early morning hours (1800 LT–0600 LT). Besides, the enhanced amplitude of BC concentration at Kolkata compared to Delhi during the late evening is also due to the increased accumulation of BC pollutants owing to the land-breeze activity during winter (Verma et al., 2016)."

6. The captions are too long of most of the figures. If possible, please short it to better readability.

Response: We have checked the captions and the required changes have been included.

7. At few places' sentences are bit long and complex so these long sentences should be broken into small sentences which will be helpful for readers

Response: We have checked throughout the manuscript for any long sentence and revised them accordingly.

8. Though authors have discussed their findings but still I feel more discussion about BC concentrations over IGP region.

Response: Based on the Reviewer's suggestion, more discussion about BC concentrations and their transport over the IGP has been included (lines 439–448). We have included more discussion on BC transport over the high-altitude location, Nainital (lines 429–433). An explanation comparing BC concentration (daytime and all-day mean) between Delhi and Kolkata has also been included (lines 457–467).

9. The conclusion section should be short and crispy for better readability.

Response: This is done. We have re-looked at the Conclusion and revised it.

**Referee 5:**

**General comment:** This work estimated radiative impact of black carbon (BC) aerosols over one of the most polluted region in the world, IndoGangetic Plain (IGP) using a high resolution ($0.1° \times 0.1°$) chemical transport model simulation. Authors assessed the five BC emission inventories (Constrained, bottomup - Smog, Cmip, Edgar, and PKU) and through a detailed statistical analysis of simulated BC concentration with respect to measurements, and constraint the large uncertainty of direct radiative forcing (RF) of BC. My specific observations are following, hope these could help improve the manuscript and make it better suitable for publishing in ACP.

Response:Thank you for sparing time reading our manuscript. The comments and suggestions have been helpful to improve our manuscript. Please find the replies to specific comments herewith:

1. Since it is an offline modeling study, only direct RF can be assessed but not indirect RF. This point deserves a further stress in the title and abstract. Also, I think some discussion regarding the potential influence (direct/indirect adjustment) of BC on cloud and cloud associated RF is worth to be commented in the manuscript.

Response: We have slightly modified the title of paper as "Wintertime direct radiative effects due to black carbon (BC) over Indo-Gangetic Plain as modelled with new BC emission inventories in CHIMERE".

Required changes have been made in the manuscript. We have also acknowledged in the 'Introduction' regarding the focus of the present paper on aerosol–radiation interactions only and presenting the direct radiative effects due to BC. Please see Section 1, lines 106–111. We have also included some more details related to WRF-CHIMERE modelling system to account for the aerosol indirect effects. Please see changes included in Section 2.1, lines 126–129. The included lines are also mentioned below:

Section 1, Lines 106–111:
"Note that applications presented in this paper focus on the aerosol–radiation interactions only and show the wintertime direct radiative perturbations or the direct radiative effects (DRE) due to BC. The study of indirect aerosol effects referring to cloud–aerosol interactions, evaluating changes in the number of cloud condensation nuclei, including the perturbations of the cloud albedo and rainfall (Boucher et al., 2013; Lohmann and Feichter, 2005) is currently ongoing and shall be presented in a forthcoming study. "

Section 2.1, Lines 126–129:
"Further, to compute the radiative perturbations due to BC, an offline coupling is again executed, forcing the WRF model with aerosol optical properties computed from CHIMERE output (refer to Section 2.3). Thereby implying the need to incorporate interactions between the two models using a WRF-CHIMERE online coupled modelling system for computing aerosol-radiation-cloud interactions (Briant et al., 2017; Péré et al., 2011)."

2. Details of each BC Emission inventory is missing. More details are appreciated here. Such as what emission sectors are considered, how seasonal variation are considered (only Dec. is simulated) and etc.

Response: Necessary changes have been included in the revised manuscript. Please see Section 2.1.3, lines 190–208, Section 2.1, lines 133–136. Please also see modified Table 1, which now includes more details about BC emission inventory. The included lines are also given below:

Section 2.1.3, Lines 190–208
"The classified source sectors of BC emission from the emission inventory database include residential, open burning, energy and industry, and transportation. The annual BC emission strength corresponding to each of the source sectors is also mentioned in Table 1 (top). The fuel combustion activity among the source sectors includes the combustion of fuelwood, crop-waste, dung-cake, kerosene, and cooking-LPG for residential cooking and heating corresponding to the 'residential' sector; open burning of agricultural residue, grassland, trash, and forest biomass to 'open burning' source sector; coal, and diesel for energy to 'energy and industry' sector; and diesel, petrol, gasoline to the 'transportation' sector. Based on the available information on sector-wise BC emission source strength (Table 1 (top)), the residential sector is seen to be the largest contributor to BC emission over the Indian region consistent with Venkataraman et al. (2005). The magnitude of annual BC emission source strength corresponding to all the sectors except the 'energy and industry' sector is estimated to be 2 to 3-times larger for the constrained emission than the bottom-up. This is specifically larger for the open burning sector, noted as 3-times the bottom-up

Smog-India, thereby suggesting the specific improvement required in quantifying the BC emission strength of the open burning sector in the bottom-up BC emission inventory. Interestingly, compared to the rest of the other source sectors, the BC emission source strength of the 'energy and industry' sector from the constrained emission matches relatively well with that from the bottom-up Smog-India. The seasonality in the spatial and temporal distribution of BC emission strength is inferred mainly from the open burning sector due to region- and season-specific prevalence of open burning of the crop residues after harvesting of Rabi or Kharif crops, including that of forest biomass burning over the Indian subcontinent (Venkataraman et al., 2006; Verma et al., 2017). The BC emission flux is also noted as being the largest during winter months over the entire Indian subcontinent and is specifically large over the IGP (Verma et al., 2017)."

Section 2.1, Lines 133–136
Evaluation of atmospheric BC concentration and BC-AOD in the present study is done during the winter month of December when the winter season is well developed in India and when the monthly mean of BC concentration is typically observed being the highest (e.g. Pani and Verma, 2014).

3. Could you explain what do you mean by "constrained emission inventory"? Do you mean top-down based on satellite observation? More details about this is also required in the method.

Response: This is done. More details about estimation of constrained emissions are provided in Section 2.1.3, lines 172–180, (also given below):

" The constrained BC emissions or so-called observationally-constrained BC emissions were estimated using integrated forward and receptor modelling approaches (Kumar et al., 2018; Verma et al., 2017). The estimation was done extracting information on initial bottom-up BC emissions and atmospheric BC concentration from the general circulation model (Laboratoire de Météorologie Dynamique atmospheric General Circulation Model (LMDZT-GCM)) simulation. The receptor modelling approach involved estimating the spatial distribution of potential emission source fields of BC based on mapping concentration weighted trajectory (CWT) fields of measured BC (daytime averaged) at the identified stations over the Indian region. The constrained BC emissions were then obtained, modifying the initial or baseline bottom-up BC emissions of the GCM corresponding to the emission source fields of BC, constraining the simulated BC concentration in the GCM with the observed BC (refer to Verma et al. (2017) for formulation and details)."

4. Maybe I overlooked, but, it seems authors forgot to describe the AERONET AAOD dataset in the Method section.

Response: This is done. Information on AERONET observations used in the study is now also provided in Table 2. A brief description on AERONET AAOD dataset is now included in Section 2.2, lines 280–289. The included lines are also given below:

"Further, the aerosol optical depth due to BC (BC-AOD) estimated in the present study (refer to Section 2.3) is compared with aerosol absorption AOD (AAOD) from Aerosol Robotic Network (AERONET; level 2) based measurements over the IGP (Giles et al., 2012; Holben et al., 1998) at stations of Kanpur, New Delhi-IMD, Gandhi College (25.87°N, 84.12°E) and IIT Kharagpur extension at Kolkata. The wintertime AAOD available from AERONET observations for Kanpur, New Delhi-IMD, Gandhi College (December 2010–2015 averaged) and Kolkata (February 2009 averaged) are used in the comparison. For Kolkata, the comparison is also made with the estimated BC-AOD of the December 2010 period as obtained from the configured aerosol model using in-situ ground-based observations of the same period (Verma et al., 2013). The AERONET AAOD data are available at four wavelengths: 440, 675, 870, and 1020 nm. The AAOD at the wavelength of 550 nm (used for comparison with simulated BC-AOD in the present study) is obtained based on the wavelength dependence of AAOD as per Giles et al. (2012)."

5. line-159: SO2 is not aerosol

Response: We have now corrected this, please see line 209:

"Besides BC emission, emission of OC, $SO_2$, and PPMr are also implemented in CHIMERE.".

6. As best as I know, the latest EDGAR emission is with 0.1° ×0.1° . resolution. Please double check and correct it accordingly in Table 1.

Response: Yes, Please find the correction in Table 1

7. As best as I know, WRF model is km-based, not deg.-based, so how could possible to perform WRF simulation in 0.1° ×0.1° resolution?

Response: Yes, you are right, WRF internal calculations are meters-based. But for the domain definition, you can use either km or lon-lat projection. In our case, we selected a regular lon-lat projection. WRF then read these longitudes/latitudes and internally convert it to run in meters. The projection used is now more detailed in the manuscript. Please see Section 2.3, lines 312–314. The modified lines in the manuscript are given below:

"Aerosol radiative transfer calculations are done in WRF-solar at a temporal resolution of 1 hour and horizontal grid resolution of 0.1° × 0.1° selecting a regular longitude-latitude projection. The WRF Preprocessing System (WPS) internally converts grid resolution corresponding to longitude-latitude projection in the degrees to meters required for model processing."

8. I think more validation of model results would be needed. I not sure if it is appropriate to validate the model (2015 simulation) with observations in about 10 years ago (2004. . .2011, in Table 2). If most of observations are available in 2004, why not perform simulation in 2004, what is your specific reason to simulate 2015? And, only validate the surface meteo. may not be enough, some validation of vertical profiles would be appreciated, since radiative transfer is a vertical path.

Response: We have modified the mentioned details in Section 2.2 regarding the observational data for clarity. Usage of observational data from available studies, which belong to measurements during different years at stations over IGP, is acknowledged in the manuscript (refer to Section 2.2, lines 248–260). As suggested, we have now also evaluated the vertical distribution of potential temperature as obtained from WRF simulation with that from measurements at a station from an available study. This is now mentioned in Section 2.2, lines 237–240. Please also refer to Figure 2(m) and the discussion in Section 3.1, lines 271–275 in the revised manuscript.

The new and revised lines included are also given below:

Section 2.2, Lines 241–246:
To compare simulated BC surface concentration with observations, measured BC surface concentration is obtained at stations over the IGP from available studies (refer to Table 2 and references therein). The selected stations correspond to area types identified as megacity (Delhi and Kolkata), urban (Agra, Kanpur, Prayagraj (or Allahabad) and Varanasi), semi-urban (Kharagpur, Ranchi, and Bhubaneshwar) and low polluted (Nainital). Comparing model results with measurements thus aids in fulfilling the requirement to evaluate the model performance towards reproducing the observed spatial patterns in BC distribution for the various area types.

Section 3.2, Lines 394–397
The simulated spatial pattern (refer to Figures 3f-g) of BC surface concentration while exhibiting the lowest value at high altitude and low-polluted location (e.g., Nainital), and the moderately high values at semi-urban stations (e.g., Kharagpur and Ranchi) is seen to reach the maximum at megacities (Kolkata and Delhi).

The simulated spatial pattern is consistent with observations, as the features of BC distribution for the specific area types are represented well by the simulated BC distribution.

Section 2.2, Lines 248–260:
It is to be noted that observational data used for comparing model estimates (meteorological parameters and BC aerosols), belong to measurement during different years at stations over the IGP. The inter-annual variability of PBLH (based on observations over Delhi) is reported as within 10% (Iyer and Raj, 2013)) and of surface temperature (based on available measurements data of India Meteorological Department over Kolkata and Kharagpur) is less than 6%. The inter-annual variability of atmospheric BC concentration over the Indian subcontinent is obtained as 5%–10% (Safai et al., 2014; Surendran et al., 2013; Bisht et al., 2015; Ram et al., 2010; Kanawade et al., 2014; Pani and Verma, 2014). Taking into account that the reported inter-annual variability of meteorological parameters

and atmospheric BC concentration is nearly equivalent to or within the uncertainty range for measurements and also is much lower than the discrepancy between simulated and observed BC as reported in previous studies (refer to Section 1). The uncertainty range is taken into consideration while evaluating the model performance compared to measurements. The comparison between model estimates and measurements at widespread geographical locations and area types as presented in this study is, therefore, justified and is primarily required for evaluating the model performance.

Section 2.1, Lines 135–136
Simulation is done for the year 2015 as the recent bottom-up BC emission database over India implemented for the year 2015.

Section 2.2, Lines 237–240
"The vertical distribution of the wintertime monthly mean of the potential temperature (Stull, 1988) as obtained from WRF simulated temperature is also compared with the measured vertical distribution of potential temperature in December (obtained for 2-days) at a station of Kanpur from an available study (Table 2) corresponding to the overlapping time hours (1000–1200 LT) from measurements."

Section 3.1, Lines 371–375
"The vertical profile of potential temperature (Figure 2m) from WRF (wintertime monthly mean) resembles well with that from measured at a station of Kanpur, with the bias being less than 4% up to the height of 500 m and less than 1% at a higher altitude (> 500 m). "

9. Why December is chosen for the simulation, any specific reason? I see in the introduction, authors mention that BC concentration is high in winter IGP, but this high concentration is with respect to surface concentration, however, this paper focuses on radiative forcing which is a vertical column transfer path. High surface pollutants concentrations in winter IGP could be solely due to shallow PBL in winter compared to summer (Chen et al., 2020), but not necessarily a high column loading which is more relevant to RF. Some comments of this could help strengthen the discussion.
Reference Chen, Y., Wild, O., Conibear, L., Ran, L., He, J., Wang, L., and Wang, Y.: Local characteristics of and exposure to fine particulate matter (PM2.5) in four indian megacities, Atmospheric Environment: X, 5, 100052, https://doi.org/10.1016/j.aeaoa.2019.100052, 2020.

Response: Required changes have been done in the revised manuscript. Please see Section 2.1, lines 133–136; and Section 1, lines 101–106. Please also refer to Section 3.3 which includes analysis of 
[revised manuscript text omitted]

---

## Author Response (AR2)

**Response to Reviews of manuscript entitled "Wintertime direct radiative effects due to black carbon (BC) over Indo-Gangetic Plain as modelled with new BC emission inventories in CHIMERE (ACP-2020-511)"**

We thank the Editor for their valuable comments, suggestions, and corrections. Specific changes made in response to the comments are described below.

**Response to Editor's comments**

There are still a few remaining issues, as detailed below. Please further address, clarify and revise.

Although claimed "This is done", it seems that

(1) no additional quantitative results/information have been provided in the abstracted - related to reviewers' comment "In the Abstract more quantitative information should be included".

Response: Additional quantitative results/information have been provided in the abstract now. The following lines have been additionally included:

"The mean BC emission flux from the five BC emission inventory database was found to be considerably high (450–1000 kg km$^{-2}$ yr$^{-1}$) over most of the IGP, with this being the highest (>2500 kg km$^{-2}$ yr$^{-1}$) over the megacities (Kolkata and Delhi)."

"The wintertime direct radiative perturbation due to BC aerosols from the *Constrained* comprised of the radiative warming at the atmosphere (+30 to +50 W m$^{-2}$) estimated to be about 50%–70% larger than the cooling due to BC at the surface."

(2) the conclusion has not been shortened or revised to be more concise, but rather lengthened - related to reviewers' comment "The conclusion section should be short and crispy for better readability".

Response: We have revised the Conclusion and tried to keep it as concise as possible. The Conclusion is shortened by about 170-words compared to the previous version. The texts in the Conclusion are required for presenting the outcomes of the manuscript adequately. Please see the uploaded "manuscript with track changes".

---

## Author Response (AR3)

**Response to Reviews of manuscript entitled "Wintertime direct radiative effects due to black carbon (BC) over Indo-Gangetic Plain as modelled with new BC emission inventories in CHIMERE (ACP-2020-511)"**

We thank the Editor for their valuable comments, suggestions, and corrections. Specific changes made in response to the comments are described below.

**Response to Editor's comments**

One remaining technical yet important issue is regarding running WRF-Chem with $0.1° \times 0.1°$ degree resolution. I double checked and found that indeed it is an option for WRF and as explained in your last response, WPS is converting the projection to km for each grid cell. But I am wondering how this dx and dy matrices are passed into the Chem module (e.g., the emission matrix is km-based, and the matrices of dx and dy are needed there)? Could you please point out the coding in the chem module that makes such simulation compatible? As there seems to be quite few studies have performed degree-based simulation with WRF-Chem, at least I haven't found any after spending some effort searching the literature, it would be very important if this issue can be clarified. Also it would be very helpful if the authors could provide some references to previous studies which also made such configurations in WRF-Chem simulations.

Response: In our study, we have used the WRF-CHIMERE modelling system.

**For CHIMERE**: We thank the Reviewer for this question. But please note that we are not using WRF-CHEM, but another modelling system, WRF and CHIMERE. As already in the manuscript, WRF can run with a grid defined either in lon-lat or meters. In this study, CHIMERE (Mailler et al., 2017) is used in offline mode. CHIMERE reads the WRF hourly meteorological fields and interpolates these meteorological fields if the CHIMERE grid is different. The interpolation is a bilinear interpolation, ensuring mass conservation for variables needing it. CHIMERE also reads anthropogenic emissions fields. The user can use the CHIMERE dedicated program (called EMISURF, see Menut et al. (2012)) or make its own program and create a file on the CHIMERE grid directly. Please see pages 105-106 in the CHIMERE user guide (`https://www.lmd.polytechnique.fr/chimere/docs/CHIMEREdoc2017.pdf`) for the user's pre-compiled emissions preparation.

The above details for CHIMERE simulation are now also included in the manuscript (refer to Section 2.1.1).

**For WRF-CHEM**: The domain definition (either km or longitude-latitude projection) is given in the control file "namelist.input" that governs the WRF or WRF-CHEM simulations. The user has to prepare the emissions in a netcdf format, suggested being compatible in format with emissions such as EDGAR_HTAP emissions database. For the WRF or WRF-CHEM setup, the WRF Preprocessing System (WPS) internally converts grid resolution corresponding to longitude-latitude projection in the degrees to meters required for model processing. The netcdf emission files created above is processed through an emissions control file to generate emissions in a model format usable by the simulation. Please see pages 12 and 26 in the WRF-CHEM user guide (`https://ruc.noaa.gov/wrf/wrf-chem/Users_guide.pdf`) for the emissions generation and use.

**References**

Mailler, S., Menut, L., Khvorostyanov, D., Valari, M., Couvidat, F., Siour, G., Turquety, S., Briant, R., Tuccella, P., Bessagnet, B., et al.: CHIMERE-2017: from urban to hemispheric chemistry-transport modeling, Geoscientific Model Development, 10, 2397–2423, https://doi.org/https://doi.org/10.5194/gmd-10-2397-2017, 2017.

Menut, L., Goussebaile, A., Bessagnet, B., Khvorostiyanov, D., and Ung, A.: Impact of realistic hourly emissions profiles on air pollutants concentrations modelled with CHIMERE, Atmos. Environ., 49, 233–244, https://doi.org/10.1016/j.atmosenv.2011.11.057, 2012.